# Electron-spin decoherence in trityl radicals in the absence and presence of microwave irradiation

Gunnar Jeschke[1], Nino Wili[1,2], Yufei Wu[1], Sergei Kuzin[1], Hugo Karas[1], Henrik Hintz[3], and Adelheid Godt[3]

[1]Department of Chemistry and Applied Biosciences, Institute of Molecular Physical Science, ETH Zurich, Vladimir-Prelog-Weg 2, 8093 Zurich, Switzerland

[2]Interdisciplinary Nanoscience Center (iNANO) and Department of Chemistry, Aarhus University, Gustav Wieds Vej 14, Aarhus C, 8000, Denmark

[3]Faculty of Chemistry and Center for Molecular Materials (CM$_2$), Bielefeld University, Universitätsstrasse 25, 33615 Bielefeld, Germany

Corresponding author: Gunnar Jeschke, gjeschke@ethz.ch

**Abstract.** Trityl radicals feature prominently as polarizing agents in solid-state dynamic nuclear polarization experiments and as spin labels in distance distribution measurements by pulsed dipolar EPR spectroscopy techniques. Electron-spin coherence lifetime is a main determinant of performance in these applications. We show that protons in these radicals contribute substantially to decoherence, although the radicals were designed with the aim of reducing proton hyperfine interaction. By spin dynamics simulations, we can trace back the nearly complete Hahn echo decay for a Finland trityl radical variant within 7 $\mu$s to the contribution from tunnelling of the 36 methyl protons in the radical core. This contribution, as well as the contribution of methylene protons in OX063 and OX071 trityl radicals, to Hahn echo decay can be predicted rather well by the previously introduced analytical pair product approximation. In contrast, predicting decoherence of electron spins dressed by a microwave field proves to be a hard problem where correlations between more than two protons contribute substantially. Cluster correlation expansion (CCE) becomes borderline numerically unstable already at order 3 at times comparable to the decoherence time $T_{2\rho}$ and cannot be applied at order 4. We introduce partial CCE that alleviates this problem and reduces computational effort at the expense of treating only part of the correlations at a particular order. Nevertheless, dressed-spin decoherence simulations for systems with more than 100 protons remain out of reach, whereas they provide only semi-quantitative predictions for 24 to 48 protons. Our experimental and simulation results indicate that solid-state magnetic resonance experiments with trityl radicals will profit from perdeuteration of the compounds.

## 1 Introduction

At sufficiently low concentration and sufficiently low temperature, coherence loss of electron spins in the absence of microwave (mw) irradiation is dominated by interaction of the electron spins with the nuclear spin bath (Mims, 1972; Zecevic et al., 1998; Soetbeer et al., 2018). Pulsed electron paramagnetic resonance (EPR) studies are often performed under such conditions in order to attain the utmost resolution for the characterization of weak interactions. About two decades ago, the quantum

information processing community started to develop approximate numerical methods for predicting electron spin decoherence caused by the nuclear spin bath (de Sousa and Das Sarma, 2003; Witzel and Das Sarma, 2006; Witzel and Sarma, 2007; Witzel et al., 2014). Later, these approaches were taken up by the EPR community (Kveder et al., 2019; Canarie et al., 2020; Bahrenberg et al., 2021; Jahn et al., 2024). Recently, we demonstrated that the nuclear spin-bath induced decay of the Hahn echo for a nitroxide radical in a water-glycerol mixture can be predicted almost quantitatively by a nuclear-pair approximation (Jeschke, 2023).

Electron-spin decoherence can be suppressed by multi-pulse sequences due to dynamical decoupling of the interaction of the electron spin with the nuclear spin bath (Witzel and Sarma, 2007; Zhang et al., 2007). This approach has found application for extending the distance range in pulsed dipolar spectroscopy experiments (Borbat et al., 2013; Spindler et al., 2015; Doll and Jeschke, 2017) and can prolong decoherence time under conditions typical in pulsed EPR by a factor of 4 to 5 (Soetbeer et al., 2018, 2021b). Decoupling of the nuclear spin bath from the electron spin can also be achieved by continuous mw irradiation (Laucht et al., 2017). For a variant of Finland trityl radical in a glassy $o$-terphenyl matrix, we recently found that the decoherence time $T_{2\rho}$ perpendicular to the mw field direction exceeds the decay time of a Hahn echo by a factor of 4.5 (Wili et al., 2020). Yet, $T_{2\rho}$ was found to be about 70 times shorter than the relaxation time $T_{1\rho}$ parallel to the mw field direction. Unlike for nitroxide radicals in the same matrix (Soetbeer et al., 2018), deuteration of the $o$-terphenyl did not lead to prolongation of the decoherence time in the absence of mw irradiation, as measured by Hahn echo decay. Protons in the trityl radical may thus cause this decay, whereas for nitroxide radicals matrix protons make the dominant contribution. As Finland trityl radicals feature twelve methyl groups, the different behavior may be caused by echo modulation induced by methyl tunnel splitting (Simenas et al., 2020). This in turn suggests that methyl-tunnel induced electron-spin decoherence (Soetbeer et al., 2021a; Eggeling et al., 2023) can be suppressed by continuous mw irradiation to some extent. However, neither the Hahn echo decay of the Finland trityl variant in $o$-terphenyl nor the decoherence of electron spins during continuous mw irradiation are presently understood. Trityl radicals are employed in dynamic nuclear polarization (DNP) schemes that involve continuous mw irradiation. Although these experiments are performed at much higher radical concentrations, understanding of the nuclear-spin contribution to $T_{2\rho}$ is of interest for optimizing such schemes.

Here we study decay of the Hahn echo and of the primary echo of dressed electron spins for three trityl radicals (Figure 1) that differ in the type and number of protons. We focus on the decay contributions from intramolecular protons that we isolate by performing the experiments in deuterated matrices. The article is organized as follows. First, we introduce the concept of the dressed spin and define the decoherence times $T_{\mathrm{m}}$ and $T_{2\rho}$. We proceed with a discussion of the spin Hamiltonian and show that methyl-tunnel induced decoherence can be treated by the recently introduced nuclear pair ESEEM formalism (Jeschke, 2023). We explain in a semi-quantitative picture why hyperfine decoupling by mw irradiation is expected to slow down echo decay caused by nuclear spins. Then we assess the suitability of the cluster correlation expansion (CCE) (de Sousa and Das Sarma, 2003; Witzel and Das Sarma, 2006; Witzel and Sarma, 2007; Witzel et al., 2014) for numerical treatment of the problem and introduce partial CCE that reduces computational effort and improves numerical stability by a well-defined truncation of the considered correlations. We continue by the analysis of experimental results and numerical computations for the three radicals. Finally, we draw some general conclusions and point to questions that remain open.

**Figure 1.** Structures of the three studied trityl radicals. The Finland trityl radical variant (FTR **1**) (trityl $CO_2H$/$CCSi^iPr_3$/$CCSi^iPr_3$) was measured in perdeuterated $o$-terphenyl and OX063 as well as OX071 in a 1:1 (v/v) mixture of either $H_2O$ and glycerol or $D_2O$ and glycerol-$d_8$. The sidegroup protons that were considered in all spin dynamics simulations are highlighted in red. Additional protons in FTR **1** that were considered in auxiliary bare-spin decoherence simulations are highlighted in violet.

## 2 Theory

### 2.1 Definition of the decoherence times $T_{\mathrm{m}}$ and $T_{2\rho}$

Decoherence of electron spins depends on temperature, electron-spin concentration, composition of the nuclear spin bath, and on the experimental scheme used for observation of the coherence evolution. In the context of this work we consider decoherence in the limit where the contribution from interactions between electron spins is negligible (low-concentration limit) and where spatial dynamics of the system does not contribute either (low-temperature limit). For observing Hahn echo decay of nitroxide radicals (Soetbeer et al., 2018) and trityl radicals (Soetbeer et al., 2021b) in protonated and deuterated matrices, these limits are attained at a concentration of 100 $\mu$M and a temperature of $40\dots50$ K (Soetbeer et al., 2018). In this work, we performed the experiments at a concentration of 100 $\mu$M and a temperature of 50 K. We define the bare-spin decoherence time (in the absence of an mw field) as $T_m$ and associate it with the decay of a Hahn echo $(\pi/2) - T/2 - (\pi) - T/2$-echo (Fig. 2a) when incrementing evolution time $T$. This is the simplest experiment that cancels the contributions to coherence dcay by a distribution of resonance offsets and by secular hyperfine couplings. In the low-concentration and low-temperature limit, decoherence as observed by a primary echo is dominated by processes in the nuclear spin bath, namely nuclear spin flip-flops caused by homonuclear couplings and admixture of tunnel states of methyl groups to the electron spin mediated by the hyperfine coupling of methyl protons (Zecevic et al., 1998; Kveder et al., 2019; Simenas et al., 2020; Jahn et al., 2022, 2024).

Using a concept introduced to quantum optics by Cohen-Tannoudji, a two-level system in a resonant electromagnetic field can be described as a dressed spin. This description is related to the rotating-frame description of magnetic resonance. The dressed spin behaves as another two-level system, whose quantization direction is the instantaneous direction of the electro-magnetic field. The level splitting is given by the amplitude of this field. This concept is useful as an analogy to the bare spin. Dressed-spin transitions can be excited by a second electromagnetic field, which is perpendicular to the first one and oscillates with a frequency that matches the amplitude $\omega_1$ of the first field. Such excitation can also be achieved by phase modulation

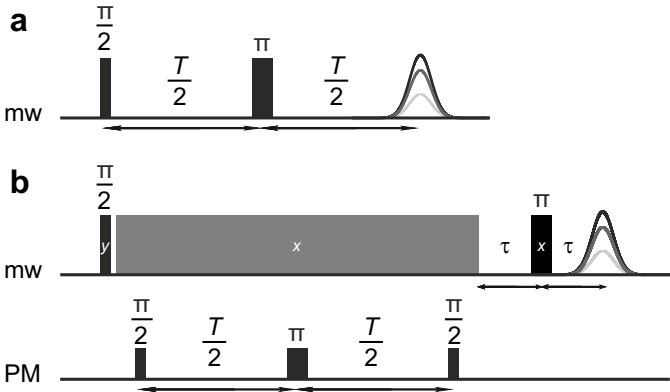

**Figure 2.** Pulse sequences for measuring bare-spin (a) and dressed-spin (b) decoherence. Thie spin-lock pulse has constant mw frequency $\omega_{\text{mw}}$ and constant amplitude $\omega_1$. The phase modulation (PM) pulses are cosine-modulated with frequency $\omega_{\text{PM}} = \omega_1$ matching the amplitude of the mw field of the spin-lock pulse (grey).

(PM) of the first electromagnetic field with a frequency matching its amplitude (Saiko et al., 2018; Wili et al., 2020). Hence, PM pulses can be assigned flip angles (Chen and Tycko, 2020) and a dressed-spin primary echo can be observed by applying

a $\pi/2 - T/2 - (\pi) - T/2 - (\pi/2)$ sequence of PM pulses during spin lock. For details on the setup of the experiment, see the Supporting Information in (Wili et al., 2020). We associate the decay of the spin-locked magnetization upon an increase in evolution time $T$ with the dressed-spin decoherence time $T_{2\rho}$. In experiments, we detect a signal proportional to the dressed-spin primary echo by stopping the spin lock followed by a detection sequence $\tau - (\pi) - \tau$-echo (Fig. 2b). In numerical simulations, we compute the expectation value of spin-locked magnetization after the second $\pi/2$ phase modulation pulse. For $T_{2\rho}$ to be

well-defined, $\omega_1$ must be much larger than the EPR linewidth for the bare spin (high-power limit). This linewidth in turn is set by the product of the static external magnetic field $B_0$ with $g$ anisotropy and the width of the hyperfine spectrum as defined in (Kuzin et al., 2022). For trityl radicals at Q-band frequencies of 34 GHz, the EPR linewidth is approximately 12 MHz. Hence, $\omega_1 = 2\pi \cdot 100$ MHz suffices. This is the mw field amplitude that we use in this work.

### 2.2 Spin Hamiltonian

We consider a single electron spin $S = 1/2$ in a nuclear spin bath consisting of $N$ proton spins $I_n = 1/2$ $(n = 1 \ldots N)$. The proton spin bath may contain $M$ methyl groups with tunnel splittings $\omega_{\text{tunnel},\mu}$ $(\mu = 1 \ldots M)$. The electron spin has a resonance offset $\Omega_S$ that is distributed due to $g$ anisotropy and unresolved hyperfine couplings to nuclei that do not significantly contribute to decoherence, such as deuterons of the matrix. For $T_{2\rho}$ measurements we assume irradiation of the electron spin by an mw field of amplitude $\omega_1$. The spin Hamiltonian for this system can be written as

$$\hat{H} = \hat{H}_S + \hat{H}_{\text{nz}} + \hat{H}_{\text{dd}} + \hat{H}_{\text{hfi}} + \hat{H}_{\text{tunnel}} + \hat{H}_{\text{mw}} , \tag{1}$$

where the contributions are the resonance offset

$$\hat{H}_S = \Omega_S \hat{S}_z \, , \tag{2}$$

the nuclear Zeeman interaction

$$\hat{H}_{\mathrm{nz}} = \omega_{\mathrm{H}} \sum_{n=1}^{N} \hat{I}_{n,z} \, , \tag{3}$$

the hyperfine interaction

$$\hat{H}_{\mathrm{hfi}} = \sum_{n=1}^{N} A_n \hat{S}_z \hat{I}_{n,z} + B_{n,x} \hat{S}_z \hat{I}_{n,x} + B_{n,y} \hat{S}_z \hat{I}_{n,y} \, , \tag{4}$$

the nuclear-nuclear dipolar interaction

$$\hat{H}_{\mathrm{dd}} = \sum_{k=1}^{N-1} \sum_{l=k+1}^{N} \omega_{\mathrm{dd},kl} \left[ \hat{I}_{k,z} \hat{I}_{l,z} - \frac{1}{4} \left( \hat{I}_k^+ \hat{I}_l^- + \hat{I}_k^- \hat{I}_l^+ \right) \right] \, , \tag{5}$$

and the methyl-tunnel interaction, which we express with the tunnel splitting $\omega_{\mathrm{tunnel}}$ as an exchange interaction between the protons of the $\mu^{\mathrm{th}}$ methyl group (Apaydin and Clough, 1968; Kveder et al., 2019),

$$\hat{H}_{\mathrm{tunnel}} = \sum_{\mu=1}^{M} \sum_{k=1}^{2} \sum_{l=k+1}^{3} -\frac{2}{3} \omega_{\mathrm{tunnel},\mu} \left( \hat{I}_{\mu,k,x} \hat{I}_{\mu,l,x} + \hat{I}_{\mu,k,y} \hat{I}_{\mu,l,y} + \hat{I}_{\mu,k,z} \hat{I}_{\mu,l,z} \right) \, . \tag{6}$$

The double indices $\mu, k$ and $\mu, l$ refer to the $k^{\mathrm{th}}$ and $l^{\mathrm{th}}$ proton in the $\mu^{\mathrm{th}}$ methyl group, respectively.

For $T_{2\rho}$ measurements we include the mw irradiation Hamiltonian

$$\hat{H}_{\mathrm{mw}} = \omega_1 \hat{S}_x \, . \tag{7}$$

We refrain from the frame transformation that simplifies the pseudo-secular part of the hyperfine interaction to $B_n \hat{S}_z \hat{I}_{n,x}$ with $B_n = \sqrt{B_{n,x}^2 + B_{n,y}^2}$. This transformation is not convenient here, as it complicates computation of the nuclear-nuclear dipole interaction. In any case, we shall include the pseudo-secular part of the hyperfine interaction only into numerical computations and skip it in our analytical expressions from here on. Further, we restrict our treatment to methyl groups with a rotation barrier that is sufficiently high to ensure $\omega_{\mathrm{tunnel},\mu} \ll \omega_1$. In expressing the methyl-tunnel interaction as pairwise exchange coupling between protons in the same methyl group, we assume that rotor-rotor coupling between methyl groups can be neglected (Jeschke, 2022) and that the high-temperature approximation applies also to the tunnel splitting. For the high tunnel barriers of the geminal methyl groups in FTR **1**, these assumptions are unproblematic.

After describing the methyl-tunnel interaction in terms of pairwise proton-proton exchange couplings, we can combine it with the nuclear dipole-dipole interaction between these protons into a nuclear-nuclear coupling term

$$\hat{H}_{\mathrm{nn},kl} = \left( \omega_{\mathrm{dd}} - \frac{2}{3} \omega_{\mathrm{tunnel},k,l} \right) \hat{I}_{k,z} \hat{I}_{l,z} - \frac{1}{4} \left( \omega_{\mathrm{dd}} + \frac{4}{3} \omega_{\mathrm{tunnel},k,l} \right) \left( \hat{I}_k^+ \hat{I}_l^- + \hat{I}_k^- \hat{I}_l^+ \right) \, , \tag{8}$$

where we have used that $\hat{I}_{k,x}\hat{I}_{l,x} + \hat{I}_{k,y}\hat{I}_{l,y} = \frac{1}{2}\left(\hat{I}_k^+\hat{I}_l^- + \hat{I}_k^-\hat{I}_l^+\right)$ and assumed that protons $k$ and $l$ belong to the same methyl group. For pairs of protons that do not belong to the same methyl group, $\hat{H}_{\mathrm{nn},kl} = \hat{H}_{\mathrm{dd},kl}$. We can drop the nuclear Zeeman interaction, as it commutes with all other terms in the spin Hamiltonian and with the initial state of the spin system. When considering only a single spin pair, we can also drop the terms with operators $\hat{I}_{k,z}\hat{I}_{l,z}$, as they have the same property. In the absence of mw irradiation, the spin Hamiltonian thus takes the form that was already treated in (Jeschke, 2023). We just need to replace the $\omega_{\mathrm{nn}}$ in this previous treatment by $\omega_{\mathrm{dd},kl} + 4\omega_{\mathrm{tunnel},\mu}/3$ if protons $k$ and $l$ are both methyl protons within the same methyl group with index $\mu$ and keep it as $\omega_{\mathrm{dd},kl}$ otherwise.

For treating bare-spin decoherence, we can thus use the nuclear pair ESEEM expression for two coupled protons that are in turn both hyperfine coupled to the electron spin. By taking into account only the secular hyperfine coupling, we found for the Hahn echo modulation due to such a proton pair (Jeschke, 2023)

$$W_{kl}(T) = 1 - \frac{3}{2}\lambda_{kl} - \frac{1}{2}\lambda_{kl}\cos\left(\omega_{\mathrm{nZQ},kl}T\right) + 2\lambda_{kl}\cos\left(\frac{1}{2}\omega_{\mathrm{nZQ},kl}T\right) \tag{9}$$

with the modulation depth

$$\lambda_{kl} = \frac{(A_k - A_l)^2\,\omega_{\mathrm{nn},kl}^2}{\left[(A_k - A_l)^2 + \omega_{\mathrm{nn},kl}^2\right]^2} = \frac{(A_k - A_l)^2\,\omega_{\mathrm{nn},kl}^2}{16\omega_{\mathrm{nZQ},kl}^4} \tag{10}$$

and the nuclear zero-quantum frequency

$$\omega_{\mathrm{nZQ},kl} = \frac{1}{2}\sqrt{(A_k - A_l)^2 + \omega_{\mathrm{nn},kl}^2}\ . \tag{11}$$

For a nitroxide radical in water-glycerol glass, we found that bare-spin decoherence in the Hahn echo experiment was predicted with very high accuracy by the product of expression (9) over all nuclear pairs $kl$. Below we shall test the quality of this analytical pair product approximation (APPA) for bare-spin decoherence of FTR **1**. Note, however, that the $\hat{I}_{k,z}\hat{I}_{l,z}$ terms of the tunnel Hamiltonian cannot be dropped in general when treating a system with more than two nuclear spins, as they do not commute with the $\left(\hat{I}_k^+\hat{I}_q^- + \hat{I}_k^-\hat{I}_q^+\right)$ for $q \neq l$. This will become important for the treatment of dressed-spin decoherence, where the APPA fails.

In general, we can simplify the relevant spin Hamiltonian to

$$\hat{H}' = \Omega_S\hat{S}_z + \sum_{n=1}^{N}A_n\hat{S}_z\hat{I}_{n,z} + \sum_{k=1}^{N-1}\sum_{l=k+1}^{N}\omega_{\mathrm{zz},kl}\hat{I}_{k,z}\hat{I}_{l,z} - \sum_{k=1}^{N-1}\sum_{l=k+1}^{N}\frac{\omega_{\mathrm{nn},kl}}{4}\left(\hat{I}_k^+\hat{I}_l^- + \hat{I}_k^-\hat{I}_l^+\right) + \omega_1\hat{S}_x\ , \tag{12}$$

where $\omega_{\mathrm{zz},kl} = \omega_{\mathrm{dd},kl} - 2\omega_{\mathrm{tunnel},\mu}/3$ if protons $k$ and $l$ are both methyl protons within the same methyl group with index $\mu$ and $\omega_{\mathrm{zz},kl} = \omega_{\mathrm{dd},kl}$ otherwise. The same argument has been put forward in (Jahn et al., 2024), which appeared during revision of our manuscript.

## 2.3  Partial diagonalization of the spin Hamiltonian for the dressed-spin case

To obtain some insight into the dressed-spin case, we decompose Hilbert space into $2^N$ two-level subspaces that correspond to a single nuclear spin configuration $M$ described by the magnetic quantum numbers of all protons, $M = \{m_I^{(n)}\}_{n=1}^N$. A nuclear

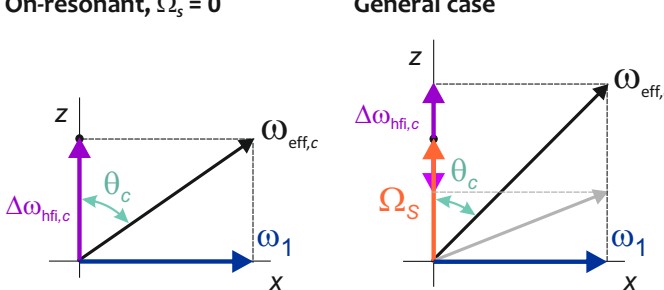

**Figure 3.** Local fields at a dressed electron spin upon on-resonant (left) and off-resonant(right) mw irradiation with amplitude $\omega_1$. In the on-resonant case, inversion of the local hyperfine field $\Delta\omega_{\mathrm{hfi},c}$ leaves the amplitude $\omega_{\mathrm{eff},c}$ of the effective field invariant, corresponding to complete hyperfine decoupling. The angle included by the effective local field and the spin-lock direction $x$ changes only sign, but not magnitude. This angle differs from zero, thus making spin-lock incomplete. In the off-resonant case, inversion of the local field (pale arrow and grey effective field) changes $\omega_{\mathrm{eff},c}$ as well as the magnitude of the angle included by the effective local field and the spin-lock direction. Hyperfine decoupling is incomplete.

spin configuration with index $c$ is characterized by a total hyperfine field

$$\Delta\omega_{\mathrm{hfi},c} = \sum_{n=1}^{N} A_n m_{I,n} \ . \tag{13}$$

The quantity $\Delta\omega_{\mathrm{hfi},c}$ corresponds to the shift of the electron spin transition frequency in subspace $c$ with respect to the resonance offset $\Omega_S$ of the uncoupled electron spin. The projection operators for the subspaces are the $\hat{P}_c = \prod_{n=1}^{N} \hat{I}^{(m_{I,n})}$, where $(m_{I,n})$ denotes $\alpha$ for $m_{I,n} = +1/2$ and $\beta$ for $m_{I,n} = -1/2$. The off-diagonal elements of $\hat{S}_x$ connect only elements within these

subspaces, but not the subspaces among themselves. For the moment, we disregard the nuclear-nuclear coupling terms in $\hat{H}'$. The Hamiltonian without these terms is diagonalized by the unitary transformation

$$\hat{U}_{\mathrm{mw}} = \prod_{c=1}^{2^N} \exp\left(-i\theta_c \hat{S}_y \hat{P}_c\right) \ , \tag{14}$$

where the tilt angles $\theta_c$ are given by

$$\theta_c = \arctan \frac{-\omega_1}{\Omega_S + \Delta\omega_{\mathrm{hfi},c}} \ . \tag{15}$$

For each nuclear spin configuration $M$, there exists a complement $\bar{M} = \{-m_I^{(n)}\}_{n=1}^{N}$ with hyperfine shift $-\Delta\omega_{\mathrm{hfi},c}$. We can combine the two subspaces to a four-level system consisting of the electron spin $S = 1/2$ and a fictitious spin $F = 1/2$. The two electron spin transitions in this four-level systems are split by

$$A_{\mathrm{eff},c} = \sqrt{(\Omega_S + \Delta\omega_{\mathrm{hfi},c})^2 + \omega_1^2} - \sqrt{(\Omega_S - \Delta\omega_{\mathrm{hfi},c})^2 + \omega_1^2} \ , \tag{16}$$

which compares to the splitting $A_c = 2\Delta\omega_{\mathrm{hfi},c}$ in the absence of mw irradiation. This reduction in level splitting is the hyperfine

decoupling effect of the mw irradiation. In general, the reduction factor differs between different nuclear spin configurations. In

the limit of on-resonant mw irradiation, $\Omega_S/\omega_1 \to 0$, we have $A_{\text{eff},c} \to 0$ for all configurations. Note, however, that, according to Eq. (15), $|\theta_c|$ differs from $\pi/2$ even for on-resonant irradiation corresponding to $\Omega_S = 0$. In other words, for all nuclear spin configurations with $\Delta\omega_{\text{hfi},c} \neq 0$ the quantization axis of the electron spin is not exactly along $x$ in the presence of mw irradiation, even if the irradiation is on-resonant. The local hyperfine field thus reduces efficiency of the spin lock. For $\omega_1 \gg$ $\Omega_S, \Delta\omega_{\text{hfi},c}$, the deviation of $|\theta_c|$ from $\pi/2$ is small, as are the differences between the $\theta_c$ of different nuclear spin configurations at the same $\Omega_S$. We have performed our measurements in this regime. For an mw field amplitude of 100 MHz and a sum of resonance offset and hyperfine field of 10 MHz, we have $90° - \theta_c < 6°$ and only about $0.5\%$ of the electron spin magnetization is not locked.

We now apply the transformation $\hat{U}_{\text{mw}}$ to the terms of the nuclear-nuclear coupling Hamiltonian. For simplicity, we first discuss the smallest spin system that exhibits all relevant phenomena. This system consists of the electron spin $S$, two coupled protons $I_1$ and $I_2$ and a bystander proton $I_3$ and features eight nuclear spin configurations. We neglect the nuclear-nuclear coupling of $I_3$ to $I_1$ and $I_2$, but consider the hyperfine coupling of $I_3$. Four of these configurations correspond to the $|\alpha_1\alpha_2\rangle$ and $|\beta_1\beta_2\rangle$ configurations of the coupled nuclear spins. The corresponding transformation operators $\hat{S}_y\hat{P}_c$ commute with the coupling operators $\hat{I}_{1,z}\hat{I}_{2,z}$ and $\hat{I}_1^+\hat{I}_2^- + \hat{I}_1^-\hat{I}_2^+$. Thus, these transformations do not affect the nuclear-nuclear coupling term. The remaining four transformations can be applied consecutively, as they pairwise commute among each other. They generate terms that connect different electron spin states or lead to different coupling between protons 1 and 2 depending on the spin state of the bystander proton. The result is conveniently expressed with tilt angle differences, $\Delta\theta_\alpha = (\theta_{\alpha\beta\alpha} - \theta_{\beta\alpha\alpha})/2$ and $\Delta\theta_\beta = (\theta_{\alpha\beta\beta} - \theta_{\beta\alpha\beta})/2$. For the off-diagonal coupling terms we find

$$
\begin{aligned}
\hat{U}_{\text{mw}} \frac{\omega_{\text{nn}}}{2} \left( \hat{I}_1^+\hat{I}_2^- + \hat{I}_1^-\hat{I}_2^+ \right) \hat{U}_{\text{mw}}^{-1} = {} & -i\omega_{\text{nn}}\hat{S}_y \left( \hat{I}_1^+\hat{I}_2^- - \hat{I}_1^-\hat{I}_2^+ \right) \left( \sin\Delta\theta_\alpha \hat{I}_3^\alpha + \sin\Delta\theta_\beta \hat{I}_3^\beta \right) \\
& + \frac{\omega_{\text{nn}}}{4} \left( \cos\Delta\theta_\alpha - \cos\Delta\theta_\beta \right) \left( \hat{I}_1^+\hat{I}_2^- + \hat{I}_1^-\hat{I}_2^+ \right) \hat{I}_3^\alpha \\
& - \frac{\omega_{\text{nn}}}{4} \left( \cos\Delta\theta_\alpha - \cos\Delta\theta_\beta \right) \left( \hat{I}_1^+\hat{I}_2^- + \hat{I}_1^-\hat{I}_2^+ \right) \hat{I}_3^\beta \\
& + \frac{\omega_{\text{nn}}}{4} \left( \cos\Delta\theta_\alpha + \cos\Delta\theta_\beta \right) \left( \hat{I}_1^+\hat{I}_2^- + \hat{I}_1^-\hat{I}_2^+ \right) .
\end{aligned}
\tag{17}
$$

We can drop the terms on the first line on the right-hand side because the $\hat{S}_y$ operator has only off-diagonal elements that connect levels that are split by about $\omega_1$. In pulsed EPR and DNP experiments, $\omega_1$ is several orders of magnitude larger than $\omega_{\text{nn}}$. The remaining terms describe a minor reduction of the nuclear-nuclear coupling between protons 1 and 2 that slightly depends on the spin state of proton 3.

This treatment can be extended by analogy to larger spin systems. Irradiation of the electron spin causes a minor reduction in the nuclear-nuclear coupling between two spins that depends on the spin states of all other nuclear spins whose hyperfine couplings are significant with respect to the mw field amplitude. For strong irradiation, the scaling factor is close to unity for the nuclear-nuclear couplings and close to zero for the hyperfine couplings. Nuclear spin pairs are most efficient in causing decoherence if the difference between their hyperfine couplings matches their nuclear-nuclear coupling. Hyperfine decoupling shifts the matching condition towards spins that have a larger hyperfine coupling in the absence of irradiation. These nuclear spins tend to be closer to the electron spin and there exist fewer of them. In addition, the scaling reduces the nuclear pair

modulation frequencies, because the hyperfine coupling contributes to these frequencies. Since the $T_{2\rho}$ measurement on dressed spins is analogous to a Hahn echo decay measurement on bare spins, we expect a weaker contribution of the nuclear spin bath on the dressed-spin decoherence time $T_{2\rho}$ than on the bare-spin decoherence time $T_{\mathrm{m}}$.

The dependence of nuclear-nuclear couplings in the vicinity of a dressed electron spin on the states of bystander nuclear spins introduces a complication. Analytical diagonalization of the spin Hamiltonian including the nuclear-nuclear coupling terms is not feasible. Hence, we shall study this case by numerical computations.

## 2.4  Initial state for $T_{2\rho}$ measurements

The initial state for $T_{1\rho}$ and $T_{2\rho}$ measurements is prepared by applying a $\pi/2$ pulse to the thermal equilibriums state of the spin system, with the latter being approximated well by a reduced density operator $\sigma_{\mathrm{eq}} = -\hat{S}_z$. With phase cycling and after normalizing signal amplitude, we obtain the initial spin-locked state that corresponds to application of an ideal pulse. For convenience, we set the phase of this pulse to $y$, so that the initial state in the rotating frame is $\sigma_{\mathrm{initial}} = -\hat{S}_x$.

In order to discuss evolution of the dressed-spin state, we need to transform $\sigma_{\mathrm{initial}}$ to the eigenbasis of the spin Hamiltonian $\hat{H}'$ given by Eq. (12). Although we cannot perform this transformation analytically, we can obtain some insight by performing the unitary transformation $\hat{U}_{\mathrm{mw}}$ defined in Eq. (14). We find

$$\sigma_{\mathrm{initial}}^{\mathrm{dressed}} = \frac{1}{2^N} \sum_{c=1}^{2^N} \sin\theta_c \hat{S}_z \hat{P}_c - \frac{1}{2^N} \sum_{c=1}^{2^N} \cos\theta_c \hat{S}_x \hat{P}_c \;. \tag{18}$$

The terms with operators $\hat{S}_x \hat{P}_c$ oscillate with frequencies that are distributed due to the distribution of resonance offsets $\Omega_S$, the different residual hyperfine shifts $\Delta\omega_{\mathrm{hfi},c}$, and inhomogeneity of the mw field that leads to a distribution of $\omega_1$. Destructive interference leads to very fast decay of these terms. Thus, we can associate $\sum_c \cos\theta_c / 2^N$ with a fast-decaying fraction of total magnetization and $f_{\mathrm{slow}} = \sum_c \sin\theta_c / 2^N$ with a long-lived fraction of magnetization. To an approximation that neglects the nuclear-nuclear couplings, we can associate $f_{\mathrm{slow}}$ with the spin-locked magnetization and its decay in the absence of phase-modulation pulses with the dressed-spin longitudinal relaxation time $T_{1,\rho}$.

Although we cannot analytically compute the second step of the transformation of the dressed-spin Hamiltonian into its eigenbasis, we can infer from the remaining off-diagonal elements that it must be of the form

$$\hat{U}_{\mathrm{nn}} = \exp\left[ -i \sum_{k=1}^{N-1} \sum_{l=k+1}^{N} \eta_{k,l} \hat{S}_z \left( \hat{I}_{k,y} \hat{I}_{l,x} - \hat{I}_{k,x} \hat{I}_{l,y} \right) \right] \;. \tag{19}$$

This transformation commutes with $\hat{S}_z$, but not with the $\hat{S}_z \hat{P}_c$ in Eq. (18). Hence, to the extent to which the $\sin\theta_c$ differ from each other, interaction with the nuclear spin bath can affect spin-locked magnetization. For $\omega_1 \gg A_n$, differences between the $\sin\theta_c$ are very small. In this regime, interference of the nuclear spin bath with the spin lock is expected to be weak. The quantization axis of the dressed electron spin depends on nuclear spin configuration $M$. However, in the regime that we discuss here, the mean magnetization along these quantization axes is close to $f_{\mathrm{slow}}$. This magnetization is spin-locked, i.e., it is a constant of motion for the dressed spin in the presence of the nuclear spin bath and in the absence of other relaxation mechanisms. In

other words, the nuclear spin bath does not lead to longitudinal relaxation of the dressed spin. The corresponding relaxation time $T_{1\rho}$ is set by other processes, such as phase noise of the mw irradiation or the density of phonons or local modes of the matrix in the vicinity of frequency $\omega_1$.

## 2.5 Cluster correlation expansion

In the discussion of Eq. (17), we have seen that the effective coupling between nuclear spins in the vicinity of a dressed electron spin is affected by the states of bystander nuclear spins. Therefore, the APPA is expected to be a worse approximation than for bare spins. As a candidate for improving on this approximation, we now consider the cluster correlation expansion (CCE) (Yang and Liu, 2008, 2009; Yang et al., 2016) that was previously applied in simulations of electron spin decoherence in dense proton baths in the absence of mw irradiation (Kveder et al., 2019; Canarie et al., 2020; Jahn et al., 2022). CCE is an attempt to systematically account for correlations among nuclear spins that contribute to electron spin decoherence. To this end, the dynamics of the entire spin bath is expanded into contributions from clusters of nuclear spins of different sizes. This expansion is truncated at a certain cluster size to balance accuracy and computational cost.

The dependence of the echo signal on time $W_c(T)$ of a cluster with $c$ nuclear spins can be computed numerically by density operator formalism. It contains contributions of all subclusters with less than $c$ spins. To obtain only the contribution of order $c$, the lower-order contributions are divided out (Yang and Liu, 2008),

$$\tilde{W}_c(T) = \frac{W_c(T)}{\prod_{\mathcal{C}' \subset \mathcal{C}} \tilde{W}_{\mathcal{C}'(T)}} \, , \tag{20}$$

where the product in the denominator runs over all subclusters. The denominator includes $W_\varnothing(T)$, which is the signal in the absence of a nuclear spin bath. For the bare-spin case, $W_\varnothing(T) \equiv 1$. For the dressed-spin case, $W_\varnothing(T)$ accounts for the time-dependent loss of magnetization due to the incomplete spin-lock, which in turn results from resonance offsets of spin packets. The contribution of all clusters of size $c$ is the product of the contributions of the individual clusters

$$\mathcal{L}_c(t) = \prod_{\mathcal{C}} \tilde{W}_c(T) \, . \tag{21}$$

Finally, the prediction of the echo decay from CCE truncated at order $o$ (CCE-$o$) is given by

$$\mathcal{L}^{(o)}(T) = W_\varnothing(T) \prod_{c \leq o} \tilde{\mathcal{L}}_c(T) \, . \tag{22}$$

For a bath with $N$ nuclear spins, CCE-$o$ thus requires computation of the $W(T)$ for $\binom{N}{o}$ clusters of size $o$, $\binom{N}{o-1}$ clusters of size $o-1$, ..., $N$ cases with a single nuclear spin, and $W_\varnothing(T)$. In order to reduce the computational effort, one can exclude clusters whose contribution one assumes to be negligible (Jahn et al., 2024). In principle, such exclusion requires an additional convergence test. For bare-spin decoherence, $\tilde{\mathcal{L}}_1(T)$ is a conventional two-pulse ESEEM signal that can be expressed as

$$\mathcal{L}_1^{\text{bare}}(T) = \prod_{k=1}^{N} W_k^{2\text{p}}(T) \, , \tag{23}$$

where $W_k^{2p}$ is the two-pulse ESEEM signal for nuclear spin $k$, whose analytical expression is known. In the high-field approximation for both the electron and nuclear spin, $W_k^{2p}(T) \equiv 1$. For dressed spins, $\tilde{\mathcal{L}}_1$ describes the interference of the hyperfine fields of all nuclei with the spin lock. If the high-field approximation does not apply to the nuclear spins, this includes magnetization loss by the NOVEL mechanism of DNP (Henstra et al., 1988; Henstra and Wenckebach, 2008). For the bare-spin case, the analytical expressions for the $W_{2,kl}$ in the high-field approximation are known as well (Jeschke, 2023). Within this approximation, APPA is identical to CCE-2.

By construction, the expansion in Eq. (22) converges to the exact signal for $o \to N$. However, CCE computation at order $N$ is more expensive than direct computation. Moreover the rate of convergence is generally unknown. Therefore, suitability of the CCE must be tested for each application. The APPA is generally deficient for dressed spins, because it fails to correct for the contribution of $\mathcal{L}_\varnothing$ to the spin-pair factors and for the modification of this contribution by $\tilde{\mathcal{L}}_1$. The approximation for dressed spins that is equivalent to the APPA for bare spins is CCE-2.

Full CCE can not be performed up to high order $o$ for two reasons. First, computation of all $\binom{N}{o}$ clusters of size $o$ is not feasible for large $o$. We find that the time for a spin dynamics computation of a cluster with $c$ nuclear spins scales as $(2^c)^{2.5}$. Taken together, for large $N$ and small $o$ this leads to an increase in the computation time by a factor of about $N$ when increasing CCE order $o$ by one. Second, as seen in Eq. (20), the computation at higher orders involves division by an increasing number of numerically computed signals. This procedure is necessarily unstable for large clusters at long times where the signal of a single cluster approaches zero. The procedure may become unstable already at smaller cluster sizes, as the numerical errors accumulate upon multiplying a large number of signals. This problem is exacerbated by the dependence of the spin dynamics simulations on the calculation of matrix exponentials and the fact that algorithms for the calculation of matrix exponentials are also approximate (Moler and Van Loan, 2003).

CCE converges quickly if the central spin, in our case the electron spin, is much more strongly coupled to the bath spins than the bath spins are coupled among themselves (Witzel et al., 2012). Hyperfine decoupling by the mw irradiation strongly reduces coupling of the central spin to the bath spins, whereas it only weakly affects coupling among the nuclear bath spins. Therefore, CCE is expected to converge more slowly for dressed-spin decoherence than for bare-spin decoherence. Strategies for improving convergence behaviour in the face of numerical instabilities have been discussed (Witzel et al., 2012). These strategies further increase computational expense. Compared to decoherence in the absence of mw irradiation (Bahrenberg et al., 2021; Jahn et al., 2024), our problem is further complicated by the longer times that we need to simulate. This is because numerical stability deteriorates with increasing evolution time.

**Table 1.** Fraction of orientations that were removed in CCE-3 computations because of numerical instabilities.

| Trityl radical | $T_m$ | $T_{2\rho}$ |
|---|---|---|
| FTR **1** | 6.0% | 46.7% |
| OX063 | 4.1% | 8.2% |
| OX071 | 0 | 5.5% |

With 24 non-exchangeable protons in OX071 and 48 such protons in OX063, the two water-soluble trityl radicals in an deuterated matrix correspond moderately-sized nuclear spin baths. For FTR **1**, we restrict CCE computations to the 36 methyl protons in the core of the radical and neglect the 42 remote protons in the two $CCSi^iPr_3$ groups.

Computational expense is then bearable up to CCE-4, but not beyond. For dressed spins, we encountered serious numerical instabilities already at CCE-3. The problem could be traced back to the Padé approximation of the matrix exponential that is used as a default in Matlab. Computation of the matrix exponential by the method of eigenvalues and eigenvectors improves numerical stability at the expense of only a slight increase in total computation time, but is insufficient for stabilizing CCE-3 for dressed spins completely. Numerical stability further improves when performing all computations in the eigenbasis of the spin Hamiltonian. However, even in this case we encounter occasional numerical instabilities in CCE-3. We treated this problem by computing powder averages with 1013 orientations and discarding signals from orientations where the simulated normalized signal at some points became negative or exceeded a value of 1.1 (see Table 1). In this, we considered only simulated data points at times shorter than the length of the experimental data trace. We found such removal necessary even for simulating bare-spin decoherence of trityls at CCE-3 level, except for OX071. Unlike for bare-spin decoherence, for dressed-spin decoherence we find strong differences between signals simulated at CCE-2 and CCE-3 level. This indicates that CCE cannot be converged for dressed spins. On the other hand, the APPA is generally deficient for dressed spins, because it fails to correct for the contribution of $W_\varnothing(T)$ and for the modification of this contribution by $\tilde{\mathcal{L}}_1$. Further, the APPA includes correlations only up to pairs and thus cannot be expected to be a good approximation in a case where CCE-2 is not a good approximation.

## 2.6 Partial cluster correlation expansion

Numerical instabilities in CCE result largely from the many Hadamard divisions of signals computed for small clusters. By partitioning of the nuclear spin bath into disjoint clusters and computing an approximation of the signal as the product of the signals of the individual clusters, this problem can be alleviated. Such cluster factorization (CF) avoids the combinatorial explosion of computation time upon increasing order $o$ of included $o$-spin correlations. This approach converges to the exact solution for $o \to N$ with less computational expense than converging CCE. On the downside, only intra-cluster correlations are included while inter-cluster correlations are neglected. The quality of the approximation thus depends on the partitioning algorithm, which should minimize inter-cluster correlations. In a recent study on bare-spin decoherence for nitroxide spin labels in a water-glycerol matrix (Jeschke, 2023), cluster factorization converged at order $o = 6$ for Carr-Purcell dynamical decoupling sequences with up to five $\pi$ pulses. However, for the case of dressed-spin coherence, exploratory simulations for single orientations in a powder average did not converge up to the largest feasible cluster size $o = 9$.

The computational expense and numerical instability of CCE at larger orders can also be tackled by disregarding clusters that are expected to make a negligible contribution. In recent work on bare-spin decoherence, two-proton clusters were disregarded at higher CCE orders if they led to negligible nuclear pair ESEEM on the timescale of the experiment according to the analytical expression for nuclear pair ESEEM (Jahn et al., 2024). For computation of dressed-spin decoherence, where the APPA performs poorly and no analytical expression is available, we do not currently have a criterion for systematically disregarding clusters. Instead, we propose to disregard part of the intercluster correlations.

Partial inclusion of intercluster correlations can be achieved by an approach that is intermediate between cluster factorization and cluster correlation expansion. For such partial cluster correlation expansion (pCCE) we partition the system into clusters of size $s$ and consider all inter-cluster correlations between $u$ of these clusters. This requires computation of $W_\varnothing$, of $N/s$ clusters of size $s$ and of $\binom{N/s}{u}$ super-clusters of size $o = us$. We denote this approach as pCCE$(s,o)$ Since the effect under consideration depends on nuclear-nuclear coupling, $s = 2$ is a safe minimum size that does not neglect any correlations. Extension to $s > 2$ somewhat resembles earlier approaches to state-space restriction (Kuprov et al., 2007). For FTR **1**, it appeared natural to us to consider the twelve individual methyl groups as strongly correlated clusters of size $s = 3$. Then it proved feasible to combine $u = 3$ methyl groups of size $s = 3$ to super-clusters of size $o = 9$, i.e., to compute decay traces at pCCE(3,9) level with 220 super-clusters. In OX063, the methyl groups are substituted by -CH$_2$-CH$_2$-OH side groups, whereas the hydroxyl protons are exchanged by solvent deuterons that are present in huge excess. Thus, we tested combining the non-exchangeable protons of each side group into a strongly coupled cluster of size $s = 4$, which is feasible up to pCCE(4,8) level with 66 superclusters. For OX071, we deal with -CD$_2$-CH$_2$-OH groups, where again the hydroxyl protons are exchanged. Here we tested combining the two side groups attached to the same ring carbon to a strongly coupled cluster of size $s = 4$, which also corresponds to pCCE(4,8). In all cases we also performed pCCE(2,6) computations and in the case of OX071 a pCCE(2,8) computation. Further work is required to find an optimal partitioning algorithm for $s > 2$.

As the strongly coupled clusters are distinct, we need to correct their simulated signals only for $W_\varnothing$, i.e.,

$$\tilde{S}_{s,j} = S_{s,j}/W_\varnothing \, , \tag{24}$$

where $S_{s,j}$ denotes the signal from the $j$-th cluster of size $s$. Inter-cluster correlations are computed by (20)

$$\tilde{\mathcal{L}}_{o,k} = \frac{S_{o,k}}{W_\varnothing \prod_{j \subset \mathcal{C}_k} \tilde{S}_{s,j}} \, , \tag{25}$$

where $j \subset \mathcal{C}_k$ now selects clusters of size $s$ that belong to the $k$-th supercluster of size $o$ that gives rise to the signal $S_{o,k}$. The pCCE signal for the whole system is given by

$$\tilde{\mathcal{L}}_o = W_\varnothing \cdot \prod_j \tilde{S}_{s,j} \cdot \prod_k \tilde{\mathcal{L}}_{o,k} \, . \tag{26}$$

Table 2 provides an overview of the different simulation approaches. In general, computation time increases dramatically with increasing correlation order and is much larger for the density-operator based CCE-2 computation as compared to the equivalent analytical computation (APPA). CCE computations can be sped up by changing the correlation type to local (Kanai et al., 2022) or by including only nuclear pairs where pair decay exceed a certain threshold (Jahn et al., 2024). In a large study, local CCE was performed with a distance cutof of 8 Å(Kanai et al., 2022). In the present work we refrained from such CCE variants, since remote interactions appear to be important for dressed-spin decoherence (vide infra).

## 3   Materials and methods

Decay curves were measured on a home-built Q-band spectrometer equipped with a Keysight M8190A arbitrary waveform generator operating at 8 GS/s, an analog-digital-converter with a sampling frequency of 2 GHz (SP Devices ADQ412), and

**Table 2.** Characteristics of the various computation approaches. Correlation order corresponds to the maximum number of nuclear spins for which correlations are included. Timing corresponds to computation of bare-spin decoherence for a single orientation of system with 36 protons (FTR **1**) on a single core of a AMD Ryzen Threadripper 3990X, 2.9-4.3 GHz.

| Approach | APPA | CCE-2 | CCE-3 | pCCE(3,9) |
|---|---|---|---|---|
| Computation type | analytical | density operator | density operator | density operator |
| Correlation order | 2 | 2 | 3 | 9 |
| Correlation type | full | full | full | partial |
| Computation time [s] | $9.5 \cdot 10^{-2}$ | 12.5 | 582 | $1.18 \cdot 10^{6}$ |

a travelling wave tube amplifier with 150 W nominal output power (Applied Systems Engineering) (Doll, 2016). By using a home-built Q-band loop-gap resonator for 1.6 mm tubes (Tschaggelar et al., 2017), we achieved a spin-lock field amplitude $\omega_1 = 2\pi \cdot 100$ MHz at a frequency of 34.8 GHz and static magnetic field of 1.2414 T (calibrated with DPPH). The temperature was stabilized at 50 K using a liquid helium flow cryostat.

The bare-spin decoherence time $T_m$ was measured with a sequence $\pi/2 - T/2 - \pi - T/2-$ echo with $t_\pi = 2t_{\pi/2} = 200$ ns (Fig. 2a). The dressed-spin decoherence time $T_{2\rho}$ was measured with the sequence in Fig. 2b by varying delay $T$ and fixed duration of the spin-lock pulse (grey) as well as fixed $\tau = 200$ ns. The spin-lock pulse of duration 35 $\mu s$ and phase $+x$ immediately followed the $\pi/2$ mw pulse of length 4 ns and phase $+y$. This spin-lock pulse had constant mw frequency and constant amplitude $\omega_1$. During PM pulses, the phase of the spin-lock pulse was cosine-modulated with frequency $\omega_{\text{mod}} = \omega_1 = 2\pi \cdot 100$ MHz. The first $\pi/2$ PM pulse of length 22 ns was applied 996 ns after the end of the $\pi/2$ mw pulse. The PM $\pi$ pulse had a length of 44 ns and the final $\pi/2$ PM pulse a length of 22 ns. PM is described by the function

$$\phi_{\text{mw}}(t) = \phi_0 + a_{\text{PM}} \cos\left(\omega_{\text{PM}} t + \phi_{\text{PM}}\right) \tag{27}$$

with the modulation amplitude $a_{\text{PM}}$, the modulation frequency $\omega_{\text{PM}} = \omega_1$ and a modulation phase $\phi_{\text{PM}}$. The $\pi$ pulse length was determined with a single PM pulse whose duration was incremented. This corresponds to a dressed-spin nutation experiment. We found that $a_{\text{PM}} = 0.3$ enabled the PM pulse lengths quoted above. Further details on experiment setup are given in (Wili et al., 2020).

For consistency with previous work (Wili et al., 2020), we used the same variant of Finland trityl radical (named FTR **1** here ) (Hintz et al., 2019) and dissolved it to a concentration of 100 $\mu$M in *ortho*-terphenyl (OTP) or its perdeuterated analogue dOTP. The perdeuterated analogue had an isotope purity of 99%, as verified by mass spectroscopy. These samples were melted with a heat gun set to 80°C and shock-frozen in liquid nitrogen before insertion into the pre-cooled resonator. We dissolved the trityl radicals OX063 and OX071 (Fig. 1) to the same concentration in water/glycerol 1:1 (v/v) or in $D_2O$/glycerol-$d_8$. These samples were shock-frozen from ambient temperature by immersion of the tube into liquid nitrogen. All samples were contained in 1.6 mm outer diameter quartz tubes.

Hyperfine tensors of the protons indicated red in Fig. 1 were computed with unrestricted Kohn-Sham density functional theory in ORCA 5.0.0 (Neese, 2022). To that end, we optimized geometry with the B3LYP functional and the D3BJ option

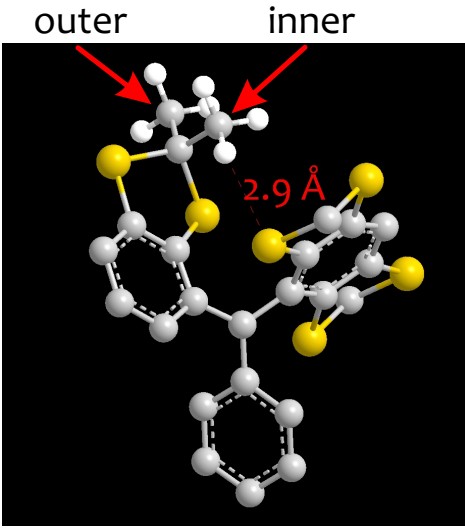

**Figure 4.** Cutout from the three-dimensional structure of the core of Finland trityl radical geometry-optimized by DFT at B3LYP/def2-SVP level. An inner methyl group with a high rotation barrier of $20.0 \pm 0.6$ kJ/mol due to interaction with a neighboring side arm and an outer methyl group with a lower rotation barrier of $15.8 \pm 0.6$ kJ/mol are indicated.

for approximating dispersion interactions, using the def2-SVP basis set for all atoms and the TightSCF option. For FTR **1**, we additionally generated a construct where the R' groups and the carboxyl group were replaced by methyl groups. Except for an APPA computation of bare-spin decoherence, all other computations for FTR **1** were performed with this smaller construct, taking into account only the 36 methyl protons of the Finland trityl core. Hyperfine tensors were obtained by a single-point computation with the B3LYP functional and the EPR-II basis set for protons and second-row atoms and the def2-TZVPPD

basis set for sulfur. Isotropic hyperfine couplings of methylene protons in OX063 and OX071 may be rather sensitive to the conformation of the sidechains. Since the 48 methylene protons in OX063 and the 24 methylene protons in OX071 sample the potential distribution of isotropic hyperfine couplings quite well, we expect minor effects from the distribution of sidechain conformations that our approach does not take into account. Nevertheless, neglect of this distribution is a potential source of disagreement between computation and experiment.

Rotation barriers for the twelve canonical methyl groups of the FTR **1** core were computed with relaxed surface scans in a 120° interval for the torsion angle in ORCA 5.0.0. We used the B3LYP functional and the D3BJ option for approximating dispersion interactions, using the def2-SVP basis set for all atoms and the TightSCF option. By fitting the obtained energies with a function $f(\phi) = V_3[\cos(3\phi + \phi_0) + 1]/2$ with variable phase $\phi_0$ we obtained the rotation barrier $V_3$ and from that the tunnel splittings $\omega_{\text{tunnel},\mu}$ as described in (Simenas et al., 2020). We found $\omega_{\text{tunnel},\mu} = 2\pi \cdot (54.5 \pm 2.6)$ kHz for outer methyl

groups and $\omega_{\text{tunnel},\mu} = 2\pi \cdot (4.8 \pm 1.8)$ kHz for inner methyl groups (see Fig. 4 for explanation). In spin dynamics simulations we assigned the mean value to all methyl groups of the same type. An analogous approach was applied for methyl groups in the $^i$Pr groups of the CCSi$^i$Pr$_3$ units. In this case we computed rotation barriers for two geminal methyl groups in a $H_3$C-Si-

$(CH-(CH_3)_2)_3$ construct. Each $^i$Pr group features an "inner" methyl group with a tunnel splitting $\omega_{\text{tunnel},\mu} = 2\pi \cdot 125.1$ kHz and an outer methyl group with a tunnel splitting of $\omega_{\text{tunnel},\mu} = 2\pi \cdot 225.1$ kHz. The methyl group at the Si atom in this construct is
sufficiently far away from the methyl groups under consideration to not influence the rotation barriers.

Prompted by a reviewer, for this smaller construct we tested a higher-level approach by employing the PBEh-3c functional, the def-TZVPP basis set, and the CPCM solvation model (Bursch et al., 2022) both for initial geometry optimization and for the relaxed surface scans. We assumed a dielectric constant of 2.5, a diffraction index of 1.62, and an effective solvation radius of 3.75 Å for $o$-terphenyl. Computation time increased by a factor of 13. The tunnel frequencies changed to $\omega_{\text{tunnel},\mu} = 2\pi \cdot 126.4$
400    kHz and $\omega_{\text{tunnel},\mu} = 2\pi \cdot 233.6$ kHz for the inner and outer methyl group, respectively. We expect other uncertainties to be much larger than this change with respect to the lower-level computation. For the larger FTR **1** construct, the computational expense of this approach is too large.

Most numerical computations were performed with EasySpin orientation grids with 9 knots (145 orientations). For pCCE(3,9) computations, we used grids with 7 knots (85 orientations). For CCE-3 computations, where we had to discard part of the ori-
entations due to numerical instability, we used grids with 23 knots (1013 orientations). Computation times were measured with the Matlab profiler. The computation time for APPA was determined by dividing the time for a powder average with 145 orientations by the number of orientations. The pCCE(3,9) computation was performed with parallelization on the level of computing the 220 combinations of three out of 12 methyl groups by using 55 cores. The time was multiplied by the number of cores.

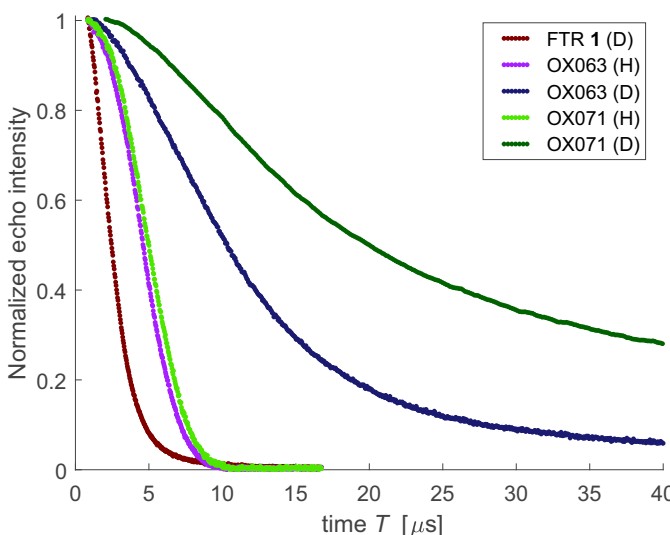

**Figure 5.** Hahn echo decays measured at 50 K and 100 $\mu$M concentration with pulse lengths of 100 and 200 ns for the $\pi/2$ and $\pi$ pulse, respectively in natural proton abundance (H) and deuterated (D) matrices for the radicals shown in Fig. 1.

# 4 Results

## 4.1 Dependence of decoherence on the type of trityl radical

The experimental results on bare-spin decoherence are summarized in Fig. 5. As reported in the Supporting Information of (Wili et al., 2020) for a compound with two Finland trityl radical units, bare-spin decoherence for FTR **1** does not depend on proton abundance in the matrix. Hence, we show data for this radical only in the perdeuterated $o$-terphenyl matrix (maroon). In contrast, deuteration of the 1:1 (v/v) water/glycerol matrix strongly reduces decoherence for OX063 (violte to dark blue) and OX071 (light to dark green) radicals, as already reported for measurements at 100 K and 10 $\mu$M concentration in (Soetbeer et al., 2021b). This can be rationalized by the similar magnitude of hyperfine couplings to the methylene protons and to matrix protons found by analysis of satellite transitions in CW EPR spectra (Trukhan et al., 2013). In the protonated matrix, the deuteration of the inner methylene groups in OX071 leads to only slight prolongation of the decoherence time, whereas in the deuterated matrix it slows down decoherence by more than a factor of two. Effects of matrix protons on bare-spin decoherence have been studied in quite some depth in recent years (Canarie et al., 2020; Bahrenberg et al., 2021; Jahn et al., 2022; Jeschke, 2023; Jahn et al., 2024) and are not expected to depend on the paramagnetic observer species. Here we focus on the effects of protons within the trityl radicals. We note that the results may be affected by incomplete deuteration of the inner methylene protons in OX071 or residual protons in the matrix.

Due to a limited gate duration of the high-power amplifier, we could measure dressed-spin primary echo decay traces only to a maximum time of 28.776 $\mu$s (Fig. 6). Hyperfine decoupling levels out the differences in decoherence behavior between the different trityl radicals as well as between protonated and deuterated matrices for the same trityl radical. It is particularly efficient for FTR **1**, where dressed-spin decoherence is only slightly faster than the one of OX063, whereas bare-spin decoherence is faster by about a factor of four. For OX071 with only 24 rather remote methylene protons, matrix deuteration substantially prolongs $T_{2\rho}$. For OX063, the 48 methylene protons make the dominant contribution to dressed-spin decoherence even in the protonated matrix, whereas the matrix dominates bare-spin decoherence. This is consistent with the expectation that hyperfine decoupling matches the (residual) hyperfine coupling difference to the nuclear-nuclear coupling for protons that are closer to the electron spin. The OX071 radical stands out by having a shorter dressed-spin decoherence time $T_{2\rho}$ than bare-spin decoherence time $T_{\mathrm{m}}$ (compare dark green and grey curves in Fig. 6).

The strong dependence of dressed-spin decoherence times on the type of trityl radical and the matrix excludes amplifier noise as the dominant source of dressed-spin decoherence. Simulations of the two experiments for a single electron spin indicated that the contributions of amplifier phase and amplitude noise to $T_{2\rho}$ are negligible at the time scales where we performed our measurements.

## 4.2 Prediction of bare-spin decoherence by the various simulation approaches

The characteristic time scale of bare-spin decoherence of FTR **1** is reasonably well predicted by any of the simulation approaches (Fig. 7), considering that DFT-predicted methyl-tunnel splittings deviate somewhat from experimental values (Simenas et al., 2020; Soetbeer et al., 2021a; Eggeling et al., 2023; Jahn et al., 2024). Due to the relatively small number of protons

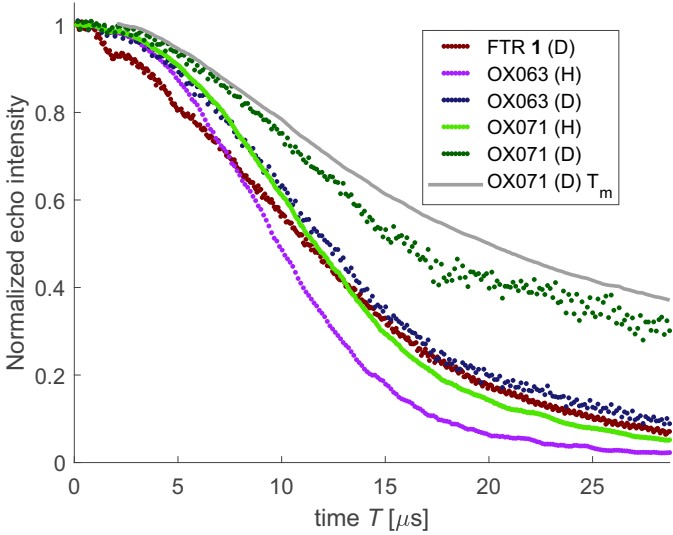

**Figure 6.** Dressed-spin primary echo decays measured at 50 K and 100 $\mu$M concentration in natural proton abundance (H) and deuterated (D) matrices for the radicals shown in Fig. 1. For comparison, bare-spin decay of OX071 (grey) is displayed as well.

and their similar magnetic parameters, the simulations show some recurrence of coherence at later times that we did not observe experimentally. With the APPA for the small construct (36 protons), recurrence is weak at the times where the experimental
445 decay trace was measured, but becomes stronger at longer times. In the CCE-3 computation (light green dots and red arrow in Fig. 7), the effect is more apparent. The strong increase at $T > 15.5$ $\mu$s for the pCCE(3,9) computation (dark green points) arises from numerical instability. With APPA for the full FTR **1** molecule (78 protons), we do not observe recurrence up to the maximum time of 40 $\mu$s for which we made the computation.

 The APPA computation with only the Finland core methyl groups (36 protons, violet) underestimates decoherence, whereas
450 the computation with all 78 non-exchangeable protons (dark blue) overestimates it. This result indicates that the "remote" methyl groups of the CCSi$^i$Pr$_3$ units contribute significantly to bare-spin decoherence. We found that the decoherence time is very sensitive to the assumed tunnel splittings, which stem from DFT computations in vacuum. Tunnel splittings in the condensed phase are likely to be smaller due to an increase of the rotation barriers from interaction with matrix molecules. For the construct with 78 protons, we can match the experimental decoherence time, though not the exact shape of the decay curve,
455 by scaling all tunnel splittings by a factor of 0.35 (maroon). Although this reduction may appear to be drastic, it corresponds to an increase of of the rotation barriers by only 1.55 kJ/mol. The different shape of the decay may result from the distribution of tunnel splittings that is seen in glassy matrices (Eggeling et al., 2023). For a model with four different types of methyl groups that reflects these features, we would need to fit mean values and standard deviations of four Gaussian distributions. We refrained from a fit of so many parameters, as we cannot expect a unique solution (Eggeling et al., 2024).

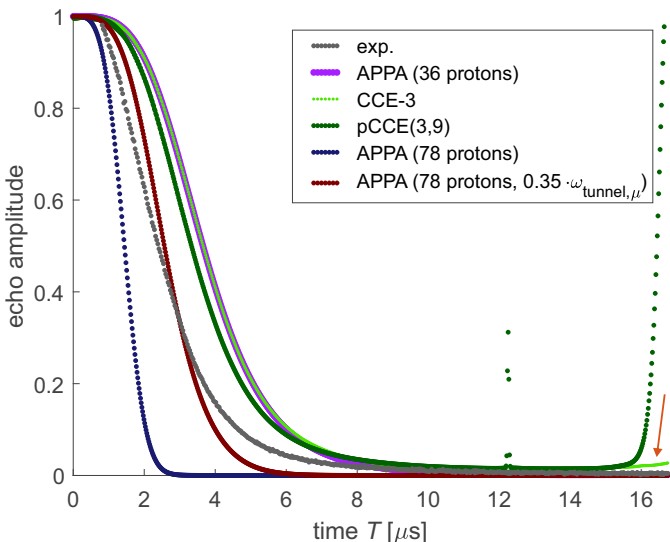

**Figure 7.** Experimental bare-spin decoherence (black) and various simulations for FTR **1** in deuterated *o*-terphenyl. The arrow points to recurrence of coherence in CCE-3.

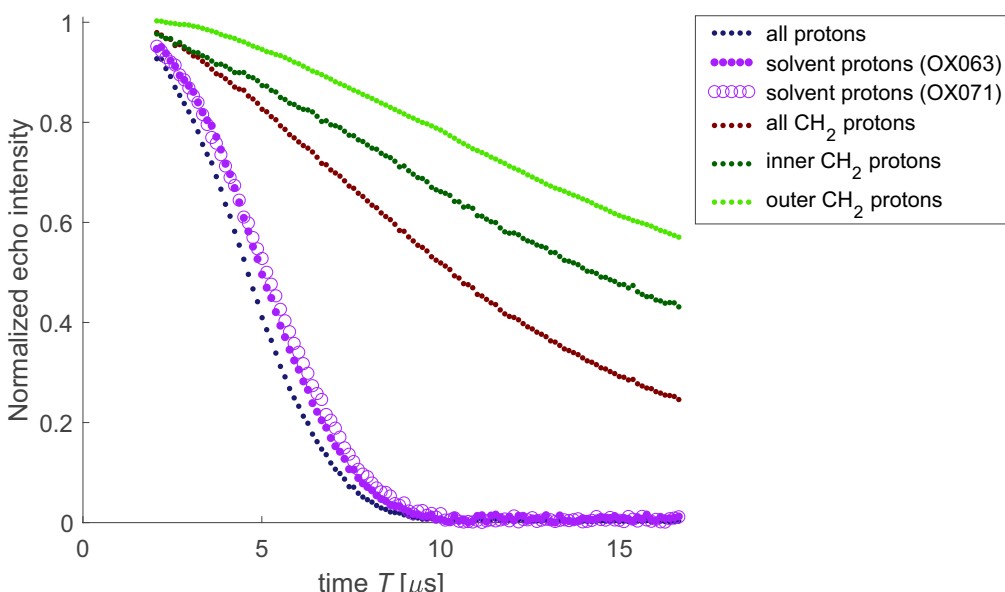

**Figure 8.** Decomposition of bare-spin decoherence contributions for the water-soluble trityl radicals OX063 and OX071. The trace for all protons (black) is the one of OX063 in protonated solvent, the one for all methylene protons the one of OX063 in deuterated solvent, and the one for outer protons the one of OX071 in deuterated solvent. The remaining traces are ratios.

Success of the APPA for the case of FTR **1** suggests that the bare-spin decoherence for OX063 and OX071 can be described as well as a product of subsets of the proton bath. Prompted by a reviewer, we applied this reasoning to a decomposition of the contributions (Fig. 8). In this approximation, the contribution of the protons in the water/glycerol matrix can be obtained by dividing the decay trace for OX063 in solvent with natural isotope abundance by the one in deuterated solvent (violet dots) or by doing the same for OX071 (open violet circles). The two estimates agree reasonably well. The contribution of all methylene

protons (maroon) is the trace for OX063 in deuterated solvent and the one for the outer methylene protons (light green) is the trace of OX071 in deuterated solvent. The contribution of the inner protons (dark green) is obtained as the ratio between the traces for all methylene protons and the inner methylene protons. As expected, the inner methylene protons contribute more strongly than the outer methylene protons.

    Turning to the simulations (Fig. 9), in the case of OX063 in deuterated water/glycerol mixture, experimental bare-spin

decoherence is faster than the one predicted by any of the simulation approaches. Although we cannot exclude residual protons in the matrix or errors in the DFT-computed proton hyperfine couplings as a reason, we note that the differences between the simulation approaches are more prominent than for FTR **1**. The APPA performs best, but this may be a case of error compensation. We do not see a reason why inclusion of additional correlations in CCE-3 (light green) as compared to APPA (violet), which is equivalent to CCE-2, should worsen the agreement. With respect to CCE-3, pCCE(4,8) (maroon) includes

higher correlations, but also neglects three-spin correlation between protons that belong to three different side groups. In the case at hand, this appears to improve the simulation, since the recurrence of coherence predicted by CCE-3 is certainly due to a deficiency of this approach. Unlike for tunnel splittings, for the hyperfine couplings of methylene protons we do not expect a broad distribution in a glassy matrix.

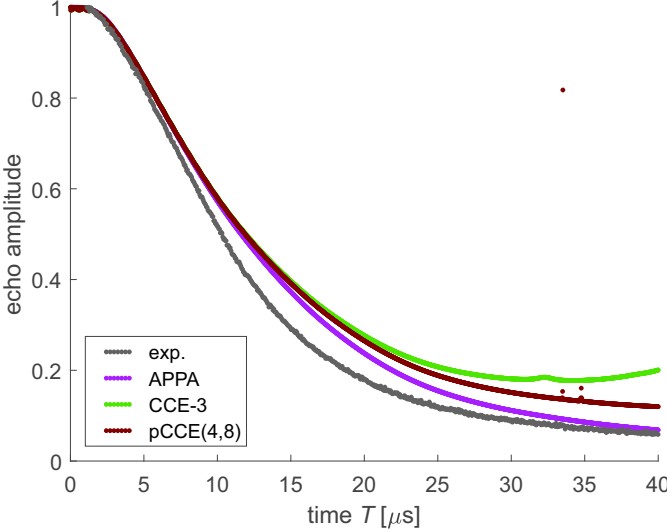

**Figure 9.** Experimental bare-spin decoherence (black) and various simulations for OX063 in deuterated water/glycerol mixture.

For OX071, we again find good agreement between the experimental bare-spin decoherence time and predictions by any of the approaches (Fig. 10). This suggests that errors in DFT-computed hyperfine couplings of the inner $CH_2$ groups in OX063 or correlations between protons of these groups rather than residual matrix protons are the reason for the deviations for OX063. For OX071, the three simulation approaches differ only at times $T > 25\mu s$.

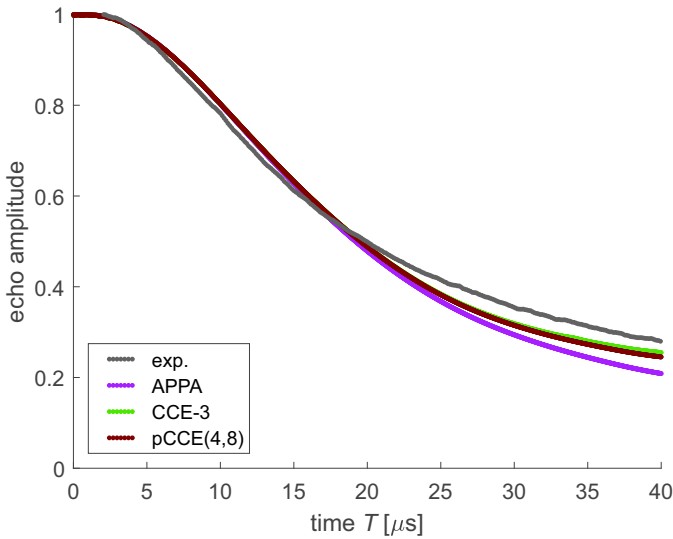

**Figure 10.** Experimental bare-spin decoherence (black) and various simulations for OX071 in deuterated water/glycerol mixture.

### 4.3 Prediction of dressed-spin decoherence by the various simulation approaches

For dressed-spin decoherence of FTR **1** (Fig. 11) we find a much larger difference between the simulation approaches than for bare-spin decoherence of the same radical (Fig. 7). This indicates that correlations between more than three spins substantially contribute to $T_{2\rho}$. The CCE-3 (light green) and pCCE(3,6) (violet) simulations differ in that CCE-3 includes correlations between three protons that reside in three different methyl groups, whereas pCCE(3,6) includes such correlations only if at least two of the three protons belong to the same methyl group. On the other hand, pCCE(3,6) includes correlations between up to six protons that belong to only two methyl groups. Despite the higher-order of included correlations, agreement with experiments at times longer than 15 $\mu s$ is worse than with CCE-3, with predicted decay being slower for pCCE(3,6). This indicates that correlations between protons in three methyl groups are significant for dressed-spin decoherence, although such protons are remote in the sense that their nuclear-nuclear coupling is very weak. In contrast, the pCCE(3,9) simulation (maroon) includes all correlations included in CCE-3 and additionally correlations of up to nine protons in up to three different methyl groups. This leads to reasonable, but not perfect agreement with experiment. We note that these computations were performed for the construct with only 36 protons of the core methyl groups, since computations with 78 protons are not feasible at this level. While we expect that, due to hyperfine decoupling, remote methyl groups contribute less to dressed-spin decoherence than

to bare-spin decoherence, we cannot test this expectation. The scatter in the pCCE(3,9)-simulated data arises from moderate numerical instability. However, we did not need to exclude individual orientations for pCCE(3,9), whereas numerical instability of CCE-3 was so serious that we had to reject the simulated signals from 46.7% of the orientations in the powder average.

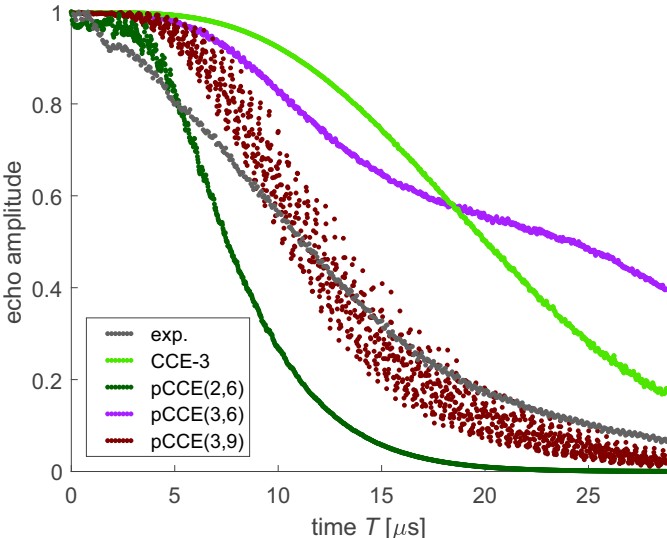

**Figure 11.** Experimental dressed-spin decoherence (black) and various simulations for FTR **1** in perdeuterated $o$-terphenyl.

The reasonable agreement of the pCCE(3,9) with experiment might be due to error compensation. This is suggested by the pCCE(2,6) simulation (dark green) predicting faster decoherence despite considering correlations to a lower order. With respect to that we note that pCCE(3,9) includes all pair correlations, i.e., any pair among the 36 protons occurs in at least one super-cluster. The slower decay of pCCE(3,9) compared to pCCE(2,6) thus implies that higher-order correlations can slow down decoherence. In order to obtain more insight, we would need to extend pCCE to higher order, which is not computationally

affordable for now.

For simulation of dressed-spin decoherence of OX063 we included a CF(4) computation, where each cluster consists of the four protons of a single -$CH_2$-$CH_2$-OD side group (dark blue). Such cluster factorization performs worse than CCE-3 (light green), again indicating that correlations between protons of different sidegroups contribute strongly to dressed-spin decoherence. Even pCCE(4,8) (maroon) performs worse than CCE-3. In contrast, a pCCE(2,6) computation (dark green)

predicts faster dressed-spin decoherence than we observe experimentally. This is in line with the observation for FTR **1** that correlations between remote protons contribute substantially to dressed-spin decoherence.

In the case of OX071, CF(4) (dark blue), CCE-3 (light green), and pCCE(4,8) (maroon) simulations strongly underestimate dressed-spin decoherence. This is similar to the behavior that we saw with the APPA and for CF of order up to 9 for Carr-Purcell dynamical decoupling sequences with an even numbers of refocusing pulses. In the case at hand, pCCE(4,8) (maroon)

performs somewhat better than CF(4) (dark blue), but worse than CCE-3 (light green). This indicates that correlations of three

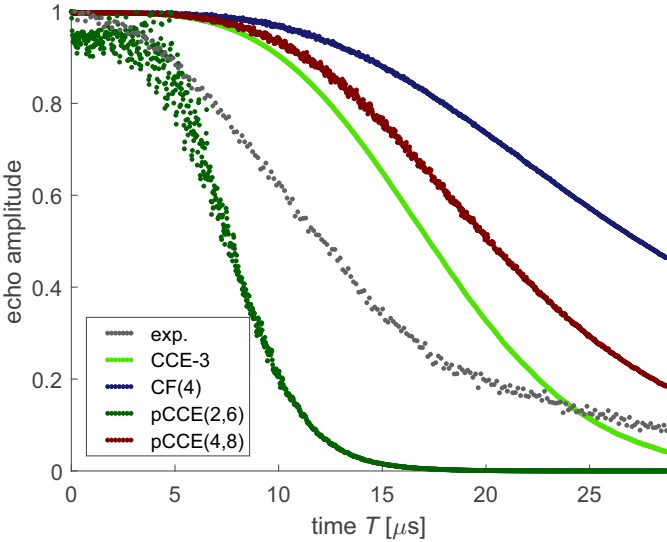

**Figure 12.** Experimental dressed-spin decoherence (black) and various simulations for OX063 in deuterated water/glycerol mixture.

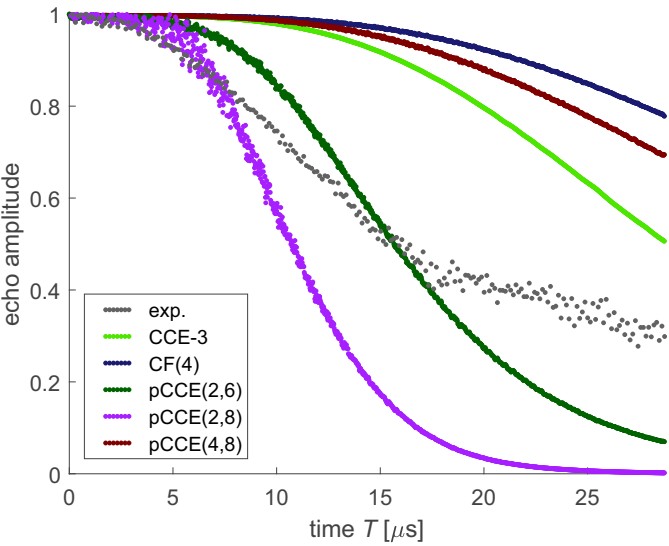

**Figure 13.** Experimental dressed-spin decoherence (black) and various simulations for OX071 in deuterated water/glycerol mixture.

protons from three different $CH_2$ groups contribute substantially to dressed-spin decoherence in OX071. We thus consistently find for all three trityl radicals that such "remote" correlations are important. By including all pair correlations at pCCE(2,6) level (dark green), we almost match the time scale of the experimentally observed dressed-spin decoherence, although the predicted decay is too slow up to a time of 15 $\mu$s and too fast afterwards. Anyway, the agreement with the time scale of the

decay might again be a result of error compensation. Extension of the highest correlation order while maintaining all pair correlations at pCCE(2,8) level (violet) leads to decay that is faster than the experimentally observed dressed-spin decoherence already at times longer than 8 $\mu$s.

## 4.4 General remarks on simulation approaches

As in previous work on nitroxide radicals in a natural proton-abundance water/glycerol matrix (Jeschke, 2023), we find that
bare-spin decoherence is adequately predicted by the APPA approach. The APPA approach is very fast and thus allows for simulations of much larger proton baths than we encountered here. We note that the APPA neglects the pseudo-secular contribution to the hyperfine coupling, whereas we included this contribution in all numerical approaches. As seen in Fig. 7, for FTR **1** the APPA agrees very well with the CCE-3 approach for the first 7 $\mu$s. At that time the coherence has fully decayed in the simulations and the approaches differ only in their recurrence behavior, which is more pronounced in the approaches
that include higher-order correlations. We note that numerical errors in treating higher-order correlations increase at longer evolution times (Witzel et al., 2012). Such errors might be the reason for the overestimate of recurrence. We cannot safely exclude, however, that some recurrence would occur even in an exact computation if all magnetic parameters and the tunnel barriers were fixed rather than distributed. That the APPA exhibits some recurrence, despite being based on analytical expressions, supports this expectation. Higher-order approaches predict slower bare-spin decoherence for OX063 and OX071 at long
evolution times than APPA. This may also be attributed to numerical errors causing some recurrence, as is clearly apparent for the CCE-3 computation for OX063 (light green dots in Fig. 9). The remaining deviations between experiment and APPA simulations of bare-spin decoherence for all three trityl radicals are more likely due to an oversimplified model of the system and errors in DFT-predicted magnetic parameters than due to neglect of higher-order correlations.

The situation is different for dressed-spin decoherence, where the APPA is not applicable and where we see more pronounced
differences between computational approaches that treat higher-order correlations in different ways (Figs. 11-13). This echoes a finding for single-nucleus ESEEM, where a product rule applies for evolution in the absence of mw irradiation, but breaks down in its presence (Jeschke and Schweiger, 1996). In the presence of an mw field, the quantization axis of the electron spin does depend on coupling to the nuclear spins. In other words, the high-field approximation breaks down for dressed electron spins, because the mw field is much lower than the static magnetic field. For the trityl radicals with 24-48 protons, CCE-3
becomes borderline numerically unstable already at shorter evolution times than we could experimentally access ($\approx 28$ $\mu$s). Computations at CCE-4 level for the FTR **1** core (36 protons) and OX071 (24 protons) exhibited grave numerical instability and could not be used. For OX063 (48 protons), we did not even attempt a CCE-4 computation because of its huge computational expense.

Higher-order correlations can be included at lower computational effort with the pCCE approach. However, this comes at
the expense of neglecting some of the correlations between remote protons at orders larger than the number $u$ of clusters that comprise a super-cluster. In the case of FTR **1**, we obtained reasonable agreement with experiment with $u = 3$. However, a pCCE(2,6) computation, also with $u = 3$, predicts too fast decay. Thus, this result should be interpreted with caution. Likewise, the pCCE(2,6) approach predicts too fast decay for OX063 and for OX071 it predicts too fast decay at long times. In the latter

case, further extension of the highest correlation order to pCCE(2,8) level worsens agreement with experiment by predicting even faster decay.

Although the pCCE approach curbs computational expense and improves numerical stability compared to the CCE approach, it is still susceptible to combinatorial explosion and to some numerical instability, as is apparent from the scatter in pCCE-simulated data. For a given experiment such scatter tends to increase with the number of super-clusters that need to be computed. Hence, for dressed-spin decoherence, where remote correlations are important and local CCE thus is not feasible, it is unrealistic to apply the CCE and pCCE approaches to systems with a much larger number of protons. This excludes computations for radicals in a matrix with natural proton abundance. As CF performs very poorly for dressed-spin decoherence, we currently do not see any approach that can provide realistic dressed-spin decoherence simulations for large and dense nuclear spin baths.

Our results on dressed-spin decoherence may shed some light on the failure of cluster factorization to converge to experimental results for Carr-Purcell dynamical decoupling with an even number of refocusing pulses (Jeschke, 2023). In these experiments, correlations between more remote protons might play a role, similar to the case of dressed-spin decoherence. Convergence of CF with respect to remote protons is expected to be much slower than convergence for vicinal protons. While pCCE computations for Carr-Purcell dynamical decoupling could shed light on this issue, they may be prohibitively expensive already at pCCE(2,4) level for a fully protonated matrix.

## 5 Conclusions

The protons in trityl radicals contribute substantially to bare-electron spin and dressed-electron spin relaxation. For FTR **1** with 12 methyl groups in the core and further 12 methyl groups in the two $CCSi^iPr_3$ substituents, this contribution causes complete bare-spin decoherence within 7 $\mu$s. For OX063 and OX071 that do not feature methyl groups, protons in a natural-abundance matrix dominate bare-spin decoherence, whereas the protons in the radicals limit coherence lifetime in deuterated matrices. These findings suggest that applications of trityl spin labels in distance distribution measurements would profit much more strongly from perdeuteration of the label than is the case for nitroxide spin labels. The same may be true for application of trityl radicals in characterization of the nuclear spin bath by the ih-RIDME approach (Kuzin et al., 2022, 2024).

Bare-spin decoherence due to the intra-radical protons in trityl radicals can be predicted quite well by the fast APPA approach (Jeschke, 2023). It remains somewhat unclear whether inclusion of higher-than-pair correlations in the much slower numerical approaches outweighs the disadvantage of the numerical errors and instabilities that these approaches exhibit at longer evolution times. Remarkably, the APPA works well for methyl-tunneling induced decoherence in FTR **1** when this effect is treated as proton exchange. Tests on different methyl-containing systems may be required before we conclude on general applicability of the APPA to methyl-tunneling induced decoherence. This approach would allow prediction of Hahn echo decay in the low-temperature and low-concentration limit from a structural model of a nanometer-sized system within a few seconds, faster even then optimized CCE-2 (Kanai et al., 2022) and corresponding to the same approximation as CCE-2. This in turn would enable the use of easily available Hahn echo decay data in refinement of ensemble models of disordered systems.

Dressed-spin decoherence is slower than bare-spin decoherence in perdeuterated matrices for FTR **1** and OX063, but not for OX071. In the latter case, experimental imperfections, such as noise of the mw source or amplifier, may play a role. We cannot exclude either that other decoherence mechanisms contribute. For instance, in previous work, a slight prolongation of Hahn echo decay upon dilution from 100 to 10 $\mu$M concentration was observed for OX063 and OX071 at a temperature of 110 K (Soetbeer et al., 2021b), suggesting a contribution by instantaneous diffusion.

We can safely conclude that protons at natural abundance in *o*-terphenyl or water/glycerol glasses make the dominating contribution to dressed-spin decoherence of trityl radicals. For deuterated matrices our simulations suggest that intra-molecular protons dominate dressed-spin decoherence for Finland trityl and OX063, and that they at least make a significant contribution for OX071. Except for FTR **1**, none of the currently available simulation approaches provides a good prediction of dressed-spin coherence. Where our simulations match the experimental dressed-spin echo decay reasonably well, we have indications that error compensation is at play. In the case of OX071, contributions to decoherence other than the one from the proton spin bath may explain part of the discrepancy. However, given the large differences between results from different simulation approaches, we anticipate that correlations between remote protons also contribute to the decoherence. We cannot draw firm conclusions on that issue at this time, as we are unable to converge the treatment of such correlations with available computational resources. Partial CCE is currently the most promising approach to this problem.

In this work, we made some progress in understanding the spin dynamics in moderately-sized electron-nuclear spin systems during mw irradiation. Most important, we find that contributions from the proton spin bath explain the time scale of such decoherence. In the near future, further understanding is unlikely to come from spending larger computational resources. Instead, we propose to study in more detail which correlations can be neglected or treated by computationally less expensive approximations.

*Code and data availability.* Experimental data, processing scripts, and simulation scripts in MATLAB are available online. DOI: 10.5281//zenodo.13850793

*Author contributions.* GJ and NW designed the research. NW carried out all measurements. YW implemented the approach for CCE simulations and for simulations of dressed-spin decoherence in general. GJ sped up the CCE code and implemented pCCE simulations. YW and GJ performed simulations and assessment of numerical stability. SK advised on treatment of the spin bath in terms of proton configurations. HK advised on improving numerical stability of propagator-based computations and performed microwave noise measurements. HH synthesized FTR **1** under the supervision of AG. The manuscript was written by GJ and edited by YW, NW, SK, HH, and AG.

*Competing interests.* The authors declare that they have no conflict of interest.

*Acknowledgements.*   This work was supported by SNSF grant 200020_219332. We thank Jan Henrik Ardankjær-Larsen for providing OX063 and OX071 and Herbert Zimmermann for providing perdeuterated *o*-terphenyl.

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
