# Peer review of "Electron-spin decoherence in trityl radicals in the absence and presence of microwave irradiation"

_Magnetic Resonance, 2024_

## Author Response (AR1)

**Reviewer comments are marked blue, response is marked black. Changes are marked red.**

Stefan Stoll, 19 Oct 2024

This is a really interesting and well-written manuscript!

It convincingly demonstrates that methyl groups in trityl radicals are not innocent bystanders when it comes to decoherence. This insight has important practical implications. Also, I find the detailed investigation of dressed-spin decoherence dynamics illuminating.

The topological partitioning used in the pCCE (partial CCE) approach is interesting and appears to work well for the trityl methyl and methylene groups, but I wonder whether the partitioning can be automated for general structures. It reminds me somewhat of Kuprov's work on Liouville space reductions, see e.g. Fig.5 in his 2007 JMR paper https://doi.org/10.1016/j.jmr.2007.09.014. On a more general level, I would expect a partitioning or clustering criterion based on interaction energies to be more efficient and generalizable than a geometry-based one.

Regarding pCCE, a cluster size of $s = 2$ does not involve any approximation for the problem at hand. This is what we meant by stating that it is "safe". We now explicitly write that it does not neglect any correlations. In this context, we also mention that extension to $s > 2$ somewhat resembles Kuprov's approach. In the case at hand, the geometric clustering criterion does correspond to an interaction-energy criterion, because the correlations arise from proton-proton dipolar couplings. Nevertheless, as you point out, our current approach is specific to trityl radicals and generalization will require additional work. We now mention this.

I am wondering whether it is possible to pinpoint the specific clusters responsible for the impressive decoherence slowdown of OX071 when going from H to D matrix (see Fig. 5). That could lead some additional valuable physical insight.

In the current work, we deliberately focus our numerical computations to protons in the trityl radicals. Matrix effects were treated by work of your own group (that we now cite in the context of Figure 5) as well as in our work on nuclear pair ESEEM. They should not depend on the paramagnetic observer species.

Regarding the methylene hyperfine couplings in OX071 and OX063 (mentioned in line 391), how strong is the isotropic contribution? Dihedral angle variations could modulate the isotropic part substantially.

The mean ratio between the absolute values of the isotropic hyperfine coupling and dipolar hyperfine coupling in OX063 is 0.16 with a standard deviation of 0.20. For OX071, these values are 0.11 and 0.13, respectively. The large standard deviations do result from different placement of the protons with respect to the main electron density in the singly occupied molecular orbital. Because of the relatively large number of methylene protons, we expect only minor effects from dihedral angle variations, but we cannot exclude such effects. We now mention this in the Materials & Methods section.

In line 470 it is stated that it is unrealistic to apply CCE and pCCE to systems with much larger number of protons (compared to 36 presumably). My lab has run CCE on systems with up to about 1000 protons (see Canarie et al, JPCL 2020; Bahrenberg et al, MR 2022; Jahn et al, JPCL 2022, all cited in this manuscript; Jahn et al, JCP 2024 in press). We even have managed to run CCE-6, although only on our HPC facility.

Line 470 is in a paragraph that discusses dressed-spin decoherence. In this case, "local CCE", which reduces the number of clusters by proximity arguments or by arguments on the decay contribution from a proton pair, is not expected to work well. Full CCE is not feasible, as is also mentioned in Jahn et al, JCP 2024, which has meanwhile appeared. We now explain our remark in more detail and have also inserted a paragraph about feasibility of full and local CCE in section 2.6 where we introduce pCCE. Per request of the two anonymous reviewers, we also included a table (Table 2) in section 2.6 that compares the approaches. The discussion of this table mentions local CCE. We also explain why we did not apply it in the context of dressed-spin decoherence.

Line 494: The vision that decoherence can be calculated from a structural model in a short amount of time has been realized with CCE-2 by Kanai et al in PNAS 2022 https://doi.org/10.1073/pnas.2121808119 on 12000 structures, see also the Python package PyCCE by Onizhuk et al.

494: We now cite the Kanai et al. paper in this context, which uses local CCE-2 with a maximum nuclear-nuclear distance of 8 Å. However, we hold that the analytical pair product approximation is still (much) faster to compute than CCE-2. For bare-spin decoherence due to nuclear pair interactions, the result is identical to the one of CCE-2. The contribution from single-nucleus ESEEM (if needed) can be computed by an analytical expression as well, with the product rule being exact for this contribution. We have slightly extended the section on CCE to discuss this. New Table 2 also explains our preference for the APPA over equivalent CCE-2.

Small comments:

136: It appears that instead of "factorize" it would be better to write "decompose". A product of 2^N two-level spaces would give 2^(2^N) states.

136: "Decompose" is indeed better than "factorize", thanks. Changed.

164: Just to clarify here: the coupling between the bystander spin and the two other nuclear spins is neglected here, correct?

164: Indeed, only the hyperfine coupling of the bystander spin is considered and the coupling to the two other nuclear spins is neglected. We added a sentence.

330: The distinction between outer and inner methyl groups is not immediately clear from the figure.

330: We now provide a ball&stick model of a cutout of the structure, which is hopefully more clear.

**Anonymous Referee #1, 21 Oct 2024**

The article by Jeschke et al. discusses the dephasing mechanism in the bare and dressed state. It was a great read, and the article fits perfectly MR. While I have enjoyed it, I still have found it at times difficult to read. I have a few comments to mostly improve the readability of manuscript.

The dephasing models discussed are non-trivial, but the authors assume a familiarity with them. Although the spin system and Hamiltonian are thoroughly described, the text introduces the CCE model abruptly, assuming the APPA model is well-known to the reader. I suggest a minor revision to the theory section by briefly reintroducing the CCE model and providing a clearer explanation of Equation 17, without adding excessive detail. Including a figure to visually explain the pCCE approach would also be beneficial, especially since the criteria for determining cluster size are not clearly defined. Similarly, the APPA model could be reintroduced with a brief explanation, helping readers quickly grasp the distinctions between models. It would also be valuable to discuss the assumptions behind these models and how they may fail under certain conditions. Overall, I recommend adding a figure or table to summarize and compare the key aspects of the models for clarity.

We now briefly introduce the APPA model (equations 9-11 and text around them). We have rewritten the section on CCE to convey the main concepts better and to explain the relation between APPA and CCE-2 better. We added Table 2 and a short discussion of it to clarify key aspects of the models.

L204: negelct -> neglect

Corrected

Figure 7 is difficult hard to "read". I would either use lines for the theoretical model or various symbol shapes for the different theoretical models

Figure 7 is of the same type as Figures 5, 6 and 8-12. We merely forgot to replace the single point in the legend by a series of points. We have improved the legend. We also changed the color scheme to get better contrast in a color blind simulator, as requested by Copernicus.

L375: why where the simulations done in vacuum? A solvation model like COSMO may have improved the results and give more accurate tunnel splittings. More importantly, the basis is rather small which may induce errors. PBEh-3c would have been a better approach (yet fast) see 10.1002/anie.202205735 This would justify the scaling mentioned subsequently.

We thank the reviewer for pointing us to this computational methodology that we will apply in the future to methyl groups in amino acid sidechains. In the case at hand (FTR **1**), the solvent o-terphenyl has low permittivity ($\varepsilon = 2.5$) and the molecule has low polarity. Therefore, we did not invoke the CPCM model in ORCA, given the uncertainty in setting the solvation radius for oterphenyl and the expectation that steric interaction with the solvent, which is not captured by COSMO or CPCM, should be the dominating effect. With the def2-SVP basis set and the B3LYP density function and D3BJ correction for dispersion, a relaxed surface scan for methyl rotation takes 6 days computation time with a single core of a Intel(R) Core(TM) i7-9700K CPU @ 3.60GHz. We computed 12 of these. For the $^i$Pr groups of the CCSi$^i$Pr$_3$ units, we now compared our original computation to a computation at PBEh-3c/def2-TZVPP/CPCM level. On our original level, each of the two relaxed surface scans for this smaller construct took about 4 hours computation time on a single core. With the new approach, each scan took about 3 days. The rotation potential changes from 14.5 to 14.6 kJ/mol for the more restricted methyl group and from 13.6 to 13.5 kJ/mol for the less restricted group. This corresponds to changes of the tunneling frequency from 125 to 121 kHz for the more restricted group and from 225 to 234 kHz for the less restricted group. We added this information to the Materials & Methods section. It is not obvious how we should construct a smaller molecular model for FTR **1** that has the same methyl rotation barriers as the whole molecule. As the computational expense at the size of FTR **1** is too large, we cannot use the higher-level approach in this case. We now explain this.

**Anonymous Referee #2, 22 Oct 2024**

This paper is a significant milestone on the road to understanding spin decoherence in organic matrices with many protons and even methyl groups. Spin decoherence has a significant impact on many forms of magnetic resonance including pulse dipolar spectroscopies such as DEER and RIDME, pulse hyperfine spectroscopy like ENDOR, EPR-based quantum computing, and dynamic nuclear polarization, etc. Spin decoherence limits the ability to control and manipulate spins which directly impacts sensitivity and resolution in techniques that are used in many different fields. Being able to tune and control decoherence will make existing techniques more useful and can enable development of capabilities that do not exist at present.

This paper considers decoherence in a set of three trityl radicals with a range of potential decoherence routes in both proton-containing and deuterated matrices. Perhaps more significantly, it considers both the decoherence of 'bare' spins in the absence of a microwave field and of 'dressed' spins in the presence of a large resonant microwave field.

Decoherence of bare spins is simpler to treat. As the paper notes in more than one place, the different decoherence routes should be independent of each other, basically parallel kinetic pathways. When two pathways are operative in a sample, the decoherence should be the product of the decoherence of the individual paths. This expectation might be confirmed by the bare spin experiments summarized in Fig. 5. FTR has a rapid decoherence, almost entirely due to methyl group tunneling. That route is absent in OX063 and OX071 which have much slower decoherence dominated by matrix protons in the (H) samples. When the matrix proton route is removed by deuteration of the matrix, decoherence again slows to values dominated by the methylene groups in OX071 (D) or the hydroxy-ethyl groups in OX063 (D). Fig. 5 provides an excellent opportunity to demonstrate the multiplicativity of decoherence via the different kinetic pathways for decoherence of bare spins. It can be done by taking simple ratios of decays curves to get the contributions from different routes. Such a demonstration would emphasize the understanding of bare state decoherence and the difference between bare and dressed states.

In fact, the different pathways for bare spins are not strictly independent, as we have discussed in (Jeschke, 2023). The product rule is only an approximation for bare-spin decoherence. However, as the approximation turned out to be good in all cases that we tested to date, we followed the advice of the reviewer and added a figure (new Fig. 8). We also added a short paragraph on decomposing the data for OX063 and OX071 into contributions from the solvent and from the inner and outer methylene protons of the water-soluble trityl radicals.

Some readers may not expect solvent protons to have a larger effect on decoherence than protons on the radical in OX063 and OX071, but Hyperfine interactions of narrow-line trityl radical with solvent molecules, J. Magn. Reson., 233 (2013) 29-36. https://doi.org/10.1016/j.jmr.2013.04.017 showed that the matrix protons in Finland trityl had hyperfine couplings and spin-flip satellite line intensities comparable to those of the radical protons, consistent with a significant role for matrix protons in decoherence. The detailed dynamics of the nuclear spins determines the relative impact of matrix or radical protons on decoherence for bare spins.

Thanks for alerting us to this paper, which we now cite in the context of Figure 5.

In the case of dressed states, the total decoherence is not the result of kinetic paths operating in parallel, but of a more complicated mechanism involving all possible routes without simple relationships between the decoherence of different samples. The unfortunate result is that you can't see if you predict or calculate decoherence from a single route without also correctly calculating decoherence from all routes. This seems to be illustrated in Fig. 6 where the seems to be little relation between decoherences of all of the samples and appears to reappear frequently in the rest of the paper. The rest of the paper is focused on identifying practical methods for numerically calculating decoherence. Some computational approaches clearly diverge and do not appear promising. However, no method clearly succeeds in predicting the kinetics and timescale of decoherence in a consistent manner. There are several mentions about 'cancellation of errors' and uncertain convergence. But this is not surprising at this stage of a complicated project and is related to one character of dressed spin decoherence: you can't correctly calculate the contribution to decoherence of one route until you can correctly calculate every route.

This paper is missing a potentially important type of information relevant to the dressed spins. That is the details of the sample composition. These are not so important for the bare spins because each bare spin at this concentration undergoes decoherence independent of other spins. But in the dressed spin experiments, the dressed spins have essentially the same frequency and cross relaxation can occur more rapidly between the degenerate spins. Consequently, a small population of rapidly decohering spins can cause the entire population of spins to decohere at a faster rate. So, in future experiments, several aspects of the samples need to be carefully considered. Fortunately, most can be checked to see if they are significant and then ignored if they have no impact. Some of these aspects of the sample include:

We do not see how a small population of rapidly decohering dressed electron spins could lead to faster decoherence of the entire population at the time scales that we are accessing and the concentration that we are using. The average coupling between electron spins is too slow for communication between them on this time scale.

- Isotopic composition of the solvent. Exchange with humidity in the atmosphere by the hygroscopic glycerol/water can result in impactful amounts of protons in the deuterated solvent samples. Contaminating protons can be measured in the solvent by NMR. Whether it affects decoherence can be checked by intentionally spiking a deuterated sample to double or triple its concentration of protons.

For FTR **1** in deuterated *o*-terphenyl, we do have information on isotopic purity from high-resolution mass spectroscopy (99%). This solvent does not have exchangeable protons and is not hygroscopic. We have added this information to the Materials & Methods section.

- Similarly, the isotopic purity of deuteration of the OX063 and OX071 was not noted. It is readily measured by high resolution mass spectrometry. Its relevance can be probed by adding a small amount of OX063 to a OX071 sample. Another test could be to exploit the convergent synthesis of the trityls to make a chimeric trityl with one OX063-like ring and two OX071-like rings.

We did not synthesize OX063 and OX071 ourselves and have only small amounts of these compounds. OX063 is not deuterated. For OX071, good agreement of simulations of bare-spin decoherence with experiment indicates that incomplete deuteration of either the radical or the solvent is not a substantial issue in our samples. However, we now do mention this complication in the context of Figures 5 and 6. In the discussion of bare-spin decoherence of OX063 we did already mention this issue in the original manuscript.

- Roughly 10% of the radicals have a C-13 in natural abundance having a hyperfine coupling comparable to twice the C-13 nuclear Zeeman frequency. This provides an opportunity for strong mixing of the electron and C-13 spin states (reminiscent of the 'matched pulse' for ESEEM). The matched pulse may offer a new route for decoherence of dressed spins and could operate in samples with contaminating protons whose nuclear Zeeman frequency is roughly half of the B1 field used to dress the spins.

Thanks for the hint. In our experiments, however, we are far from matching C-13 (by almost a factor of eight). Simulations of the NOVEL effect showed a rather narrow matching range at Q-band frequencies even for protons. Such matching would also cause decay of spin-locked magnetization, i.e., $T_{1\rho}$. We did not see such an effect in (Wili et al., 2020). Hence, we can safely exclude that this effect affects our results.

- In the absence of any protons, there must be some decoherence route. It may involve deuterons and their tunneling, or instantaneous diffusion, or the influence of random 'pairs' of radicals in the sample (or pairs formed with trace amounts of divalent metal ions), or dissolved molecular oxygen, or the very low-frequency phonon and vibrational modes contributing to the anomalous low-temperature properties of glassy matrices, or some other unexpected route. Its magnitude could be checked by a perdeuterated version of OX061 in deuterated solvent.

While we agree that there must be some decoherence route in the absence of protons, we did not study such a sample and currently do not have access to a perdeuterated version of OX063. We

would suspect that in such samples decoherence might be dominated by residual protons due to the limited isotope purity that you mention above. Whereas water is available at 99.9% deuteration and sample preparation could be performed in controlled atmosphere, glycerol-d8 is available only with an isotope purity of 99%. This may or may not be sufficient to suppress the nuclear-spin bath contribution to a sufficient extent for seeing other contributions.

However, these are considerations for future work building on the base reported here. I see little need to go back at this point and characterize samples that will likely not be used in future work. I do recommend that the multiplicative nature of bare spin decoherence and its lack for dressed spins, be exposed better by additions to Figs. 5 and 6 with an explanation in the text. A short table or graphic showing the relations and differences between the calculational methods would be extremely useful for readers who are not numerically inclined or that have not been following computational spin dynamics very carefully.

We added Table 2 and a short discussion as a overview over the calculational methods.

Minor Points:

- Wouldn't the omega-dd in eq. 8 be time dependent from tunnelling if the protons were on different methyl groups?

This is not the case. Tunneling can be fully described by a static Hamiltonian.

- The 2nd line in eq. 14 seems to have a + or - sign missing.

Thanks for noticing. It must be a +. We corrected this.

- It would help to refer reader at line 220 back to where it was mentioned in the paper.

In the discussion of Eq. (14). We have added this information.

**Malcolm Levitt, 28 Oct 2024**

The description of the experiment could be clearer.

Please specify the magnetic field and the microwave frequency (not just as the code word "Q-band").

We added the frequency (34.8 GHz) and magnetic field (1.2421 T).

The pulse sequence in Fig.2a is fully specified since the the relevant pulse lengths etc. are given in the Materials and Methods section. However for Fig.2b, some aspects are unclear. The spin-lock field amplitude is specified as a nutation frequency $\omega_1$ which is given as 2pi*100MHz in the Materials and Methods. However the caption states that the additional PM

pulses have a frequency that matches \omega_1. This is misleading. Presumably the caption does not mean that the pulses have a frequency \omega_1, but that the +modulation frequency+ of the microwave field is equal to \omega_1. Furthermore, the precise form and timing of the modulation should be specified. In addition, the amplitude of those pulses does not seem to be specified, at least not very clearly. It may be there somewhere, but it should be made clearer.

In the caption of Figure 2b we now write that the phase modulation (PM) pulses have a modulation frequency that matches $\omega_1$. The microwave pulse with frequency 34.8 GHz is indicated above. We now also mention that this is a cosine modulation. The amplitude results from setting the flip angles, as for other pulses in magnetic resonance. We now point the reader to the detailed setup procedure in (Wili et al., 2020) when referring to Fig. 2b. We also added the timing of the PM pulses in Materials & Methods.

Since the paper describes attempts to match simulations and experiment, it is essential that the experimental procedure is clear and unambiguous.

We agree and thank for the advice.

---

## Author Response (AR2)

**Editor comments are marked blue, response is marked black. Changes are marked red.**

**Malcolm Levitt, 20 Nov 2024**

This excellent article should be accepted for publication in Magnetic Resonance, providing that the experimental procedure is described in sufficient detail to allow reproduction by other scientists, and to allow accurate simulations to be made. As it stands, this is still not fully the case. The caption for fig.2 has been improved, which moves in the right direction. It is now clearer that the PM pulses are implemented by applying a second, cosine-modulated microwave field, 90 degrees out of phase with the main spin-locking field, which has amplitude omega1 in frequency units, and that the frequency of the cosine modulation is also omega1. However, the amplitude of the second cosine-modulated field is still not specified, and the durations of the pulses are not given either. The authors refer in several places to ref.20 for further details but this is not an acceptable way of specifying important experimental details, and it is anyway not easy to extract these essential details from ref.20. The article may be published once precise specifications for the amplitude of the orthogonal microwave field and the durations of all pulses are given.

As described in our previous rebuttal, we had already added the timing of the phase-modulation pulses to Materials & Methods. This included the durations of all pulses. It reads: "The spin-lock pulse of duration 35 μs and phase +x immediately followed the $\pi/2$ mw pulse of length 4 ns and phase +y. This spin-lock pulse had constant mw frequency and constant amplitude $\omega_1$. During PM pulses, the phase of the spin-lock pulse was cosine-modulated with frequency $\omega_{mod} = \omega_1$. The first $\pi/2$ PM pulse of length 22 ns was applied 996 ns after the end of the $\pi/2$ mw pulse. The PM $\pi$ pulse had a length of 44 ns and the final $\pi/2$ PM pulse a length of 22 ns." We do not see a reason for duplicating this information in the caption of Fig. 2.

As usual, amplitude is defined by flip angle and pulses length, or, vice versa, at fixed amplitude, PM pulse length is determined by flip angle. We now added the procedure for setup to Materials & Methods.